# EMBO *reports*

# Muskelin is a substrate adaptor of the highly regulated *Drosophila* embryonic CTLH E3 ligase

Chloe A Briney[1,2,5], Jesslyn C Henriksen[1,2,5], Chenwei Lin [ID][3], Lisa A Jones[3], Leif Benner[4], Addison B Rains[1,2], Roxana Gutierrez[1,2], Philip R Gafken[3] & Olivia S Rissland [ID][1,2 ✉]

## Abstract

The maternal-to-zygotic transition (MZT) is a conserved developmental process where the maternally-derived protein and mRNA cache is replaced with newly made zygotic gene products. We have previously shown that in *Drosophila* the deposited RNA-binding proteins ME31B, Cup, and Trailer Hitch are ubiquitylated by the CTLH E3 ligase and cleared. However, the organization and regulation of the CTLH complex remain poorly understood in flies because *Drosophila* lacks an identifiable substrate adaptor, and the mechanisms restricting the degradation of ME31B and its cofactors to the MZT are unknown. Here, we show that the developmental regulation of the CTLH complex is multi-pronged, including transcriptional control by OVO and autoinhibition of the E3 ligase. One major regulatory target is the subunit Muskelin, which we demonstrate is a substrate adaptor for the *Drosophila* CTLH complex. Finally, we find that Muskelin has few targets beyond the three known RNA-binding proteins, showing exquisite target specificity. Thus, multiple levels of integrated regulation restrict the activity of the embryonic CTLH complex to early embryogenesis, during which time it regulates three important RNA-binding proteins.

**Keywords** E3 Ligase; Maternal-to-zygotic Transition; Protein Degradation
**Subject Categories** Development; Post-translational Modifications & Proteolysis

## Introduction

The maternal-to-zygotic transition (MZT) is an essential process in early animal embryogenesis where the embryo switches from a maternally deposited mRNA and protein cache to a zygotically derived proteome and transcriptome. In mammals, the MZT occurs over the first 36–48 h of development, while in *Drosophila melanogaster* the transition occurs in the syncytial zygote over

the course of ~5 h (Vastenhouw et al, 2019). Despite timescales being vastly different between different animals, the molecular cascades that orchestrate the MZT are remarkably well-conserved. Maternally deposited RNAs are rapidly destroyed in flies, zebrafish, mice, and humans (Bashirullah et al, 1999; Svoboda et al, 2015; Sha et al, 2020), and at the same time, zygotic transcription is initiated (Lee et al, 2013; Higuchi et al, 2018; Heyn et al, 2014; Duan et al, 2021; Harrison et al, 2011; Larson et al, 2022; Schulz and Harrison, 2019). Although RNA regulation is much better understood than protein regulation in the MZT, a number of studies have highlighted the essential contribution of maternal protein decay during early embryogenesis (Zavortink et al, 2020; Cao et al, 2020; Gao et al, 2017; Dang et al, 2023; Kinterova et al, 2019; Shimuta et al, 2002; Cao et al, 2022).

One mechanism by which proteins are cleared is the ubiquitin–proteasome system (UPS). Here, a cascade of enzymes works together to create polyubiquitin chains on a target protein, marking it for degradation by the proteasome. First, an E1 ubiquitin-activating enzyme transfers a ubiquitin molecule to an E2 ubiquitin-conjugating enzyme, which then associates with an E3 ubiquitin ligase and the target, thereby catalyzing the ubiquitin transfer (Komander and Rape 2012). Target specificity is mainly orchestrated by the E3, which contains a substrate receptor domain or adaptor (Cowan and Ciulli, 2022). One prototypical example is the family of RING-type E3 ligases, which catalyze ubiquitin transfer directly from the E2 to the substrate by correctly scaffolding the substrate, catalytic modules, and other supporting members of the complex (Deshaies and Joazeiro, 2009; Rotin and Kumar, 2009).

The CTLH E3 ligase is a multi-subunit, RING-type ligase that has emerged as a vital player in the *Drosophila* MZT. Known as the GID complex in yeast, this E3 ligase is conserved from yeast to humans and has been implicated in several important processes including early embryogenesis, erythropoiesis, immune response, tumorigenesis, and intermediary metabolism (Zavortink et al, 2020; Sherpa et al, 2022; Simwela et al, 2024; McTavish et al, 2019; Salemi et al, 2017; Liu et al, 2020; Santt et al, 2008; Qiao et al, 2020; Sherpa et al, 2021; Gottemukkala et al, 2024; Yi et al, 2024). In flies, the CTLH complex targets the RNA-binding protein ME31B and its two partners, Cup and Trailer Hitch (TRAL), which together form

[1]Department of Biochemistry and Molecular Genetics, University of Colorado Anschutz Medical Campus, Aurora, CO 80045, USA. [2]RNA Bioscience Initiative, University of Colorado Anschutz Medical Campus, Aurora, CO 80045, USA. [3]Proteomics & Metabolomics Shared Resource, Fred Hutchinson Cancer Center, Seattle, WA 98109, USA. [4]Section of Developmental Genomics, Laboratory of Biochemistry and Genetics, National Institute of Diabetes and Digestive and Kidney Diseases, National Institutes of Health, Bethesda, MD 20892, USA. [5]These authors contributed equally: Chloe A Briney, Jesslyn C Henriksen. ✉E-mail: olivia.rissland@cuanschutz.edu

a post-transcriptional repressive complex and are critical for oogenesis (Wilhelm et al, 2005, 2003; Nakamura et al, 2001, 2004). All three proteins are maternally deposited in the embryo and then rapidly removed during the MZT by the ubiquitin–proteasome system via the CTLH complex (Wang et al, 2017; Zavortink et al, 2020; Cao et al, 2020). The degradation of ME31B and its partners is developmentally controlled, in part through a translational feedback loop: during oogenesis, the ME31B complex translationally represses the mRNA encoding the E2 dedicated to the CTLH complex (known as *Marie Kondo* or *Kdo*). However, this repression is alleviated upon egg activation through the action of the Pan Gu kinase, leading to Kdo production and full activation of the E3 ligase (Zavortink et al, 2020).

Much of the organization of the *Drosophila* CTLH complex has been inferred from work in yeast and humans (Fig. 1A; Qiao et al, 2020; Sherpa et al, 2021; Maitland et al, 2022). Broadly, the E3 ligase has three organizational domains: catalytic, scaffolding, and substrate recognition. The catalytic components form a dimer and interact with its E2 via a set of conserved interactions (Chrustowicz et al, 2024). In yeast, a set of scaffold components, including Gid8 (known as Houki [Hou] in *Drosophila*, see Fig. EV1A for all homologous naming) and Gid1 (RanBPM in *Drosophila*), connect the catalytic domain to the substrate adaptor. The substrate recognition domain is composed of Gid4 and Gid5, which help identify and orient the target protein. In both yeast and humans, the complex is then organized into a higher-order supramolecular structure through an additional β-propeller subunit (Gid7 in yeast) that increases target affinity and ubiquitylation (Langlois et al, 2022; Sherpa et al, 2021). Although most subunits are broadly conserved with easily identifiable orthologs, *Drosophila* notably lacks the two subunits needed for substrate recognition in yeast: Gid4 and Gid5 (Figs. 1A and EV1A; Zavortink et al, 2020).

Target recognition is an essential aspect of how protein degradation by E3s, including the CTLH complex, is regulated. For instance, in yeast, the CTLH complex utilizes separate substrate adaptors, such as Gid4, Gid10, or Gid11, depending on the cellular environment and needs of the organism (Barbulescu et al, 2024; Gottemukkala et al, 2024; Kong et al, 2021; Langlois et al, 2022; Maitland et al, 2024; Qiao et al, 2020; Sherpa et al, 2022). The situation is even more complex in humans, where different structural components, specifically the human orthologs of Gid7 (WDR26 and Muskelin), not only change the overall organization of the complex but also change substrate specificity. In particular, WDR26 plays a dual role in the human CTLH complex, functioning both in forming the supramolecular complex *and* in directly recognizing target proteins (Sherpa et al, 2021; Mohamed et al, 2021; Gross et al, 2024; Onea et al, 2022; Gottemukkala et al, 2024). Recent work has also demonstrated that Muskelin, the other Gid7 ortholog, is required to mediate CTLH supramolecular complex formation for degradation of a metabolic enzyme in humans in response to mTORC1 inhibition, although how Muskelin affects target specificity in this context is unknown (Maitland et al, 2024; Yi et al, 2024). Further, Muskelin also enables a version of the CTLH complex that allows FAM72A to act as the substrate adaptor for degradation of UNG2 during antibody maturation in human B cells (Barbulescu et al, 2024). Muskelin is proposed to be mutually exclusive with WDR26 in humans. It is both ubiquitylated as part of autoregulation by the CTLH complex and regulates a distinct subset of the proteome compared to

WDR26 (Jordan et al, 2023; Maitland et al, 2019, 2024; Yi et al, 2024). However, our work in S2 cells has found in this system that Muskelin and CG7611 (a WDR26 ortholog) are indeed mutually exclusive in binding to RanBPM (Fig. EV1B).

Intriguingly, the *Drosophila* genome lacks orthologs for the best-described substrate adaptor (*Gid4*) and even lacks an ortholog for *Gid5* (Zavortink et al, 2020). Moreover, the ortholog of *WDR26* in *Drosophila* (*CG7611*) is not required for the destruction of ME31B and its cofactors during the MZT (Zavortink et al, 2020), leaving open the question of the identity of the substrate receptor for the embryonic CTLH complex. In other words, two mysteries surround the *Drosophila* CTLH complex, particularly in the context of protein degradation during the MZT. First, its target ME31B is an essential RNA-binding protein that is broadly expressed alongside nearly all components of the CTLH complex: it has therefore been unclear what mechanisms might restrict the degradation of ME31B to the MZT. Given the notable absence of a *Gid4* ortholog, we reasoned that this mystery might be linked to a second one—the identity of the *Drosophila* CTLH substrate receptor.

Here, we describe multiple mechanisms that restrict the recognition of ME31B by the CTLH complex to the MZT beyond the previously established translational control of *Kdo*. First, at the end of the MZT, the interaction between the CTLH complex and its target proteins weakens during the MZT, and then autoregulation further reduces levels of complex components. Second, transcription of the Muskelin subunit (which is required for ME31B degradation) is tightly restricted to oogenesis. Muskelin expression is mediated by the OVO oogenesis-specific transcription factor, thus providing an explanation for why ME31B is not targeted outside the MZT. Importantly, we also demonstrate that Muskelin functions as the substrate adaptor for the CTLH complex that recognizes ME31B. We found that the embryonic CTLH complex robustly targets, at most, four proteins during the MZT; in other words, ME31B and its binding partners appear to be the critical targets of Muskelin. The complex layers of transcriptional, post-transcriptional, and post-translational regulation thus converge to restrict the activity of the embryonic CTLH complex—and the degradation of ME31B/Cup/TRAL—to a tight developmental window.

## Results

### The embryonic CTLH complex is autoregulated and degraded at the end of the maternal-to-zygotic transition

ME31B degradation in the MZT is mediated by the E2-E3 combination of *Marie Kondo* (*Kdo*) and the CTLH complex (Zavortink et al, 2020; Cao et al, 2020). A previous study examined protein degradation through and beyond the MZT (Cao et al, 2020), and we further investigated the temporal control of ME31B degradation by performing an extended time course through eight hours of development. As expected, both endogenous ME31B protein levels and overexpressed ME31B-GFP protein levels dropped quickly during the MZT (2–4 h after egg-laying), but they remained steady at the end of the MZT and beyond (Fig. 1B).

We next developed antibodies to components of the CTLH complex with an independent company (Boster, see "Methods"). We tested antibody specificity to Muskelin, RanBPM, Katazuke (Kaz), and

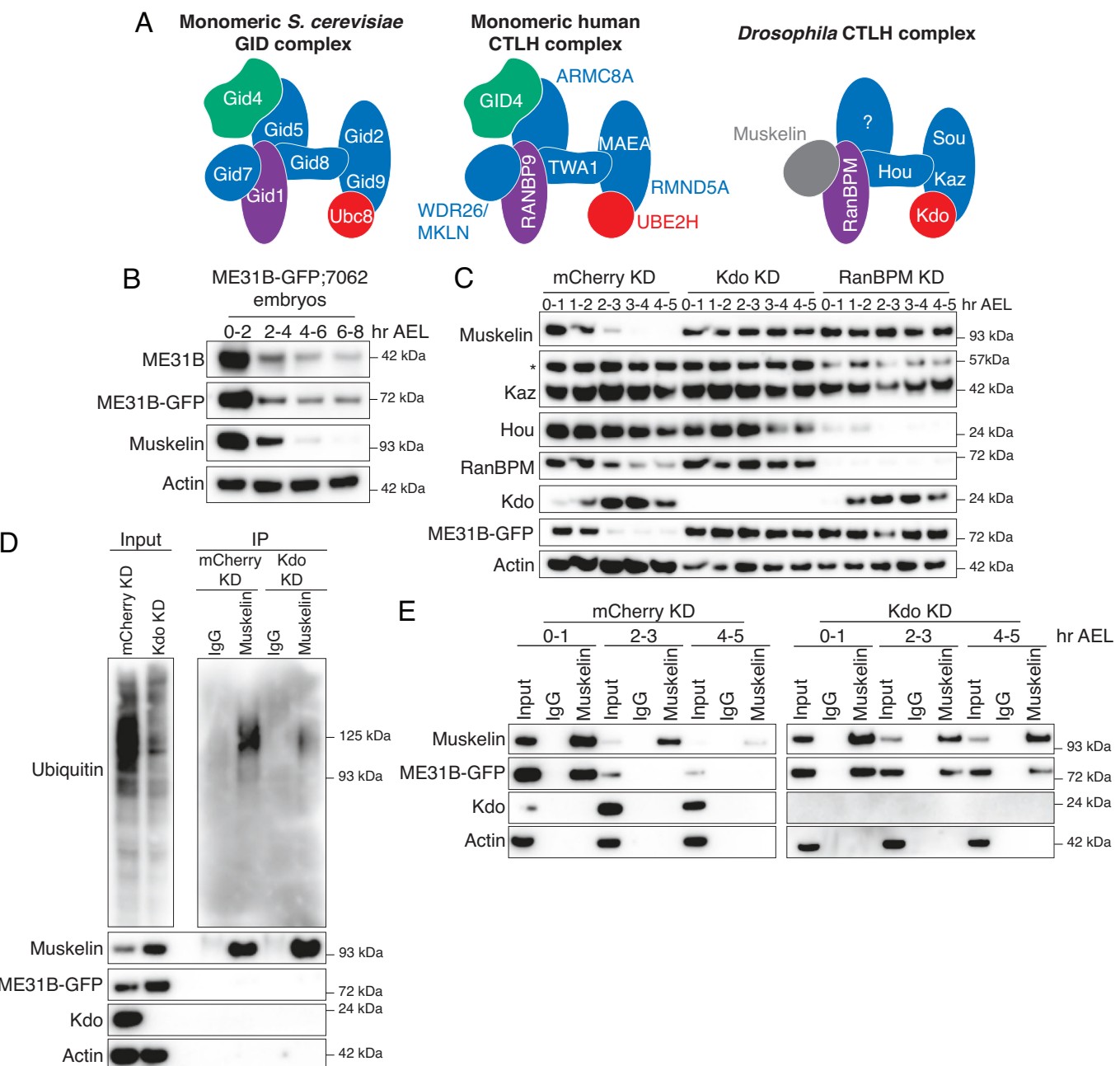

**Figure 1. The embryonic CTLH complex is autoregulated and degraded at the end of the maternal-to-zygotic transition.**

(A) Models of the CTLH complex in yeast, human, and fly. The substrate recognition module is in green, catalytic/scaffolding modules in blue, and E2 in red. (B) ME31B levels are stable after the end of the MZT. Lysates from staged embryos were probed for ME31B-GFP, ME31B, Muskelin, and actin (as a loading control) for the indicated time points. (C) The CTLH complex is autoregulated. Staged embryos from the indicated times were harvested with mCherry, Kdo, or RanBPM-depleted. Lysates were probed for the indicated proteins. (*) denotes non-specific band. (D) In the absence of Kdo, Muskelin is less ubiquitylated. Muskelin was immunoprecipitated from lysates from 1 to 2 h control (mCherry) or Kdo knockdown embryos and probed for ubiquitin as well as other indicated binding partners. (E) The interaction between Muskelin and ME31B-GFP is more robust at the end of the MZT in the absence of Kdo. Muskelin was immunoprecipitated from staged lysates across the MZT time course from control (mCherry) knockdown or Kdo knockdown embryos. Hr AEL: hours after egg-laying. All western blots are representative images from three biological replicates. Source data are available online for this figure.

Houki (Hou) using RNAi in S2 cells (Fig. EV1C) and then probed levels of Kdo and components of the CTLH complex across the MZT (Fig. 1C). Intriguingly, levels of all components, except for Kaz (one of the catalytic subunits), decreased markedly during early embryogenesis. For instance, Muskelin levels dropped soon after ME31B

degradation and were undetectable by the end of the MZT (Fig. 1B,C), followed by RanBPM, Hou, and finally Kdo itself. In other words, the subunits of the CTLH complex are additional examples of maternally deposited proteins that are degraded during the MZT, likely contributing to the temporal control of their degradation targets.

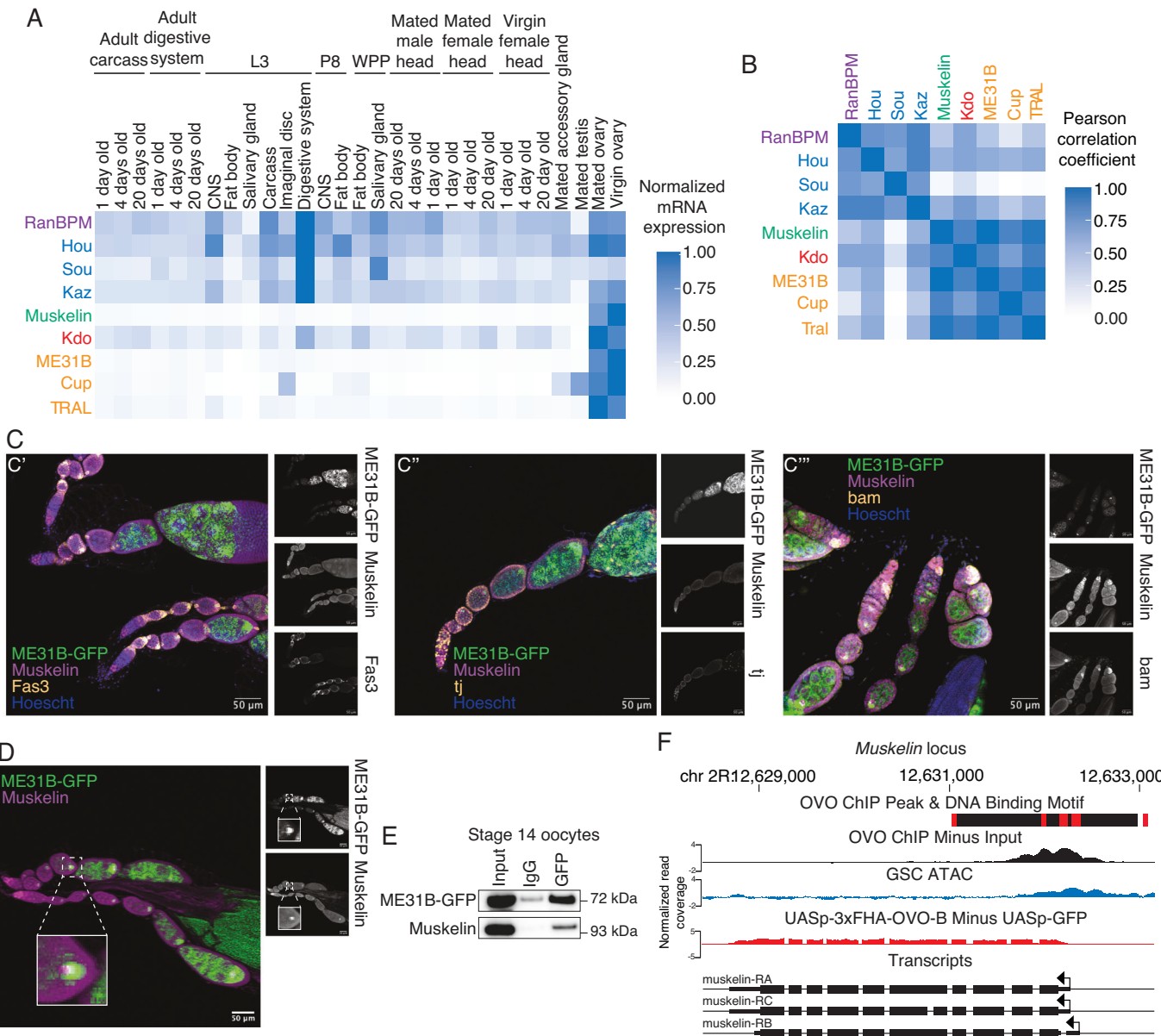

**Figure 2. Muskelin expression is restricted to oogenesis.**

(A) mRNA expression of CTLH components and targets is highest in ovarian tissues. Each gene's expression in the indicated tissue was normalized to its maximum expression. "L3": third instar larvae; "P8": pupal stage 8, "WPP": white prepupa. (B) *Muskelin* mRNA expression is most similar to *Kdo* and CTLH complex targets. Pearson correlation coefficients between absolute RPKM values across all tissues from (A) were calculated for each indicated gene pair. (C) Muskelin is expressed in a variety of cell types in the ovary. Ovariole co-immunofluorescence was performed for Muskelin (magenta) and Fas3, tj, or bam (yellow) in ME31B-GFP (green) expressing ovaries. Images are representative of at least three biological replicates. Muskelin co-localizes with Fas3 (C') and tj (C") in follicle cells and with bam (C''') in germline stem cells. Scale bar = 50 μm. (D) Muskelin and ME31B-GFP co-localize in the developing oocyte. mCherry knockdown (control) ovarioles expressing ME31B-GFP (green) were stained for Muskelin (magenta). Germline stem cells are at the left of the image with more mature egg chambers toward the right. ME31B-GFP expression is highest in the developing oocyte where co-localization with Muskelin is strongest. Scale bar = 50 μm. Inset represents a 400% magnification of the indicated oocyte. All immunofluorescence images are representative of three biological replicates. This image is the same as the image used in Fig. EV2D. (E) Muskelin and ME31B-GFP interact by immunoprecipitation in stage 14 oocytes. ME31B-GFP was immunoprecipitated from stage 14 oocyte lysate and probed for Muskelin and ME31B-GFP. Representative data from three biological replicates. (F) *Muskelin* gene level read coverage tracks for OVO ChIP minus input and GSC ATAC-seq, and *ovo^{ΔBP}/ovo^{ovo-GAL4}; UASp-3xFHA-OVO-B* minus *ovo^{ΔBP}/ovo^{ovo-GAL4}; UASp-GFP* RNAseq. Red rectangles and black rectangles represent significant OVO DNA binding motifs and OVO ChIP peaks, respectively. Gene models are represented at bottom. Small rectangles represent untranslated regions, large rectangles represent translated regions. Arrows indicate transcriptional start sites. Source data are available online for this figure.

The human ortholog of Muskelin is subject to autoubiquitylation (Maitland et al, 2019; Yi et al, 2024), and so we wondered if similar autoregulation occurred in *Drosophila*. To test this hypothesis, we knocked down Kdo or RanBPM and examined CTLH complex subunit stability throughout the MZT time course. Consistent with such a model, in the absence of the E2 Kdo, Muskelin was stabilized (Fig. 1C). In embryos depleted of RanBPM, which is predicted to connect Muskelin to the rest of the complex, Muskelin was also stabilized (Fig. 1C). Moreover, although Hou was substantially depleted in the absence of RanBPM, the residual Hou present was still degraded, again consistent with a model of autoregulation of the CTLH complex subunits (Fig. 1C). Further, previous work has shown that Muskelin is ubiquitylated during the MZT (Cao et al, 2020). We expanded on this finding by immunoprecipitating Muskelin under stringent conditions from 1 to 2 h embryo lysates; the loss of interacting proteins, such as ME31B, was confirmed by western blotting. We then probed the immunoprecipitates for ubiquitin and found that Muskelin is ubiquitylated in control embryos (Fig. 1D). The ubiquitylation signal was much weaker when Kdo has been depleted, supporting a model of auto-ubiquitylation of Muskelin by the Kdo-CTLH complex cascade.

Given that both Muskelin and ME31B levels decrease during the MZT (Fig. 1C), we next asked how their interaction changed during the same developmental window. Despite the robust reduction in levels of both ME31B and Muskelin by the end of the MZT, we were able to enrich Muskelin throughout the time course (Fig. 1E, mCherry knockdown). At the beginning of the MZT (0–1 h), we observed a robust interaction between ME31B and Muskelin by immunoprecipitation in control embryos (Fig. 1E, mCherry knockdown). In contrast, by the end of the MZT (at 4–5 h), we were no longer able to detect an interaction between Muskelin and ME31B in control embryos (Fig. 1E, mCherry knockdown). Similarly, by immunofluorescence, we observe a strong, diffuse signal for both proteins in 0–1 h embryos that robustly decreases in 4–5 h embryos (Fig. EV1D). To test whether the interaction between Muskelin and ME31B is dependent on Kdo activity, we repeated the immunoprecipitation experiment in staged embryo lysate lacking Kdo. Interestingly, when Kdo was depleted, the interaction by immunoprecipitation was now detectable even at the end of the MZT (Fig. 1E), perhaps suggesting that the decrease in levels of these proteins might partly lead to the reduced interaction. Thus, not only do the levels of the CTLH complex decrease, but also the remaining complex interacts poorly or not at all with its target ME31B during the MZT. Although not tested here, it is tempting to speculate the reduced interaction between Muskelin and ME31B may in fact trigger the autoregulation of Muskelin and the rest of CTLH complex.

## Muskelin expression is restricted to oogenesis

We next explored the developmental expression of transcripts encoding known components of the CTLH complex (*RanBPM*, *Hou*, *Sou*, *Kaz*, and *Muskelin*), the E2 *Kdo*, and its known targets (*Cup*, *TRAL*, and *ME31B*) using published RNAseq datasets from FlyBase (Fig. 2A; Öztürk-Çolak et al, 2024). Although most components showed broad expression across the available tissue atlas, they tended to be highest in the ovary. A notable exception was the component *Muskelin*, whose transcript appears to be highly restricted to the ovary, an observation we confirmed with qRT-PCR (Fig. EV2A). In fact, when we compared absolute transcript abundances of known CTLH components, *Kdo*, and CTLH targets by Pearson correlation (Fig. 2B), *Muskelin* was most similar to the *targets* of the CTLH complex, not other complex members. Similar results were seen when we analyzed published mass spectrometry and RNAseq datasets from FlyBase (Casas-Vila et al, 2017; Graveley et al, 2011); Fig. EV2B,C). Given that Muskelin is required for the degradation of ME31B, Cup, and TRAL (Zavortink et al, 2020; Cao et al, 2020), its own restricted developmental expression may limit their degradation.

These analyses of Muskelin came with two caveats, however. First, whole ovaries are a complex mix of cells. Second, even within the oocyte, complex regulatory networks can translationally repress maternal transcripts until egg activation; indeed, *Kdo* is an example of this phenomenon (Zavortink et al, 2020). In other words, the presence of an mRNA in the oocyte does not necessarily mean the presence of its encoded protein. We next explored the expression of Muskelin using immunofluorescence, probing for the endogenous protein using our newly developed antibody, which we tested for specificity using Muskelin knockdown ovaries (Fig. EV2D). In wild-type ovaries, we observed that Muskelin is present in the developing oocyte and is detectable in early germline stem cells and follicle cells (Fig. 2C). We used *bam* as a marker for germline stem cells and *Fas3* and *traffic jam* (*tj*) as markers for follicle cells Newton et al, 2015). The wide expression of Muskelin we observe by immunofluorescence is consistent with single-cell RNAseq data indicating Muskelin expression outside the germline and previous studies showing that it plays roles outside of the developing oocyte to support oogenesis (Li et al, 2022) (Kronja et al, 2014). To test whether the Muskelin knockdown flies were developing appropriately, we phenotypically analyzed whole ovaries and ovarioles dissected from control or Muskelin knockdown flies (Fig. EV2E). Further, we also performed viability studies for embryos from both knockdown conditions. Interestingly, we observe only mild phenotypic differences in ovaries or ovarioles, and there are no significant differences in F2 generation viability from Muskelin knockdown F1 parents (Fig. EV2F). While beyond the scope of our current study, we speculate that there may be compensatory changes in RNA regulation in the absence of Muskelin.

We next compared the localization of Muskelin with ME31B, using an ME31B-GFP allele expressed at the endogenous locus (Zavortink et al, 2020). We have observed that these proteins interact in the embryo (Fig. 1D), and so we were curious whether this interaction is initiated in the oocyte. Indeed, the two proteins are co-expressed in the developing oocyte (Figs. 2D and EV2D). In addition, an interaction between ME31B-GFP and Muskelin was detectable by immunoprecipitation from stage 14 oocytes (Fig. 2E). Because Kdo has not yet been produced in the oocyte (Zavortink et al, 2020), the Muskelin-ME31B interaction most likely does not result in ubiquitylation; instead, we propose that the CTLH complex is "poised" in the oocyte prior to egg activation.

Intrigued by our finding that Muskelin is expressed in germline stem cells, we considered potential transcription factors that could activate its expression. Recent work has characterized targets of the master oogenesis transcription factor OVO (Benner et al, 2024). Using data from the Benner et al, 2024 OVO study, we confirmed that importantly, the *Muskelin* promoter contains predicted OVO-binding sites and OVO ChIP-seq signal (Fig. 2F). *Muskelin* transcripts are also

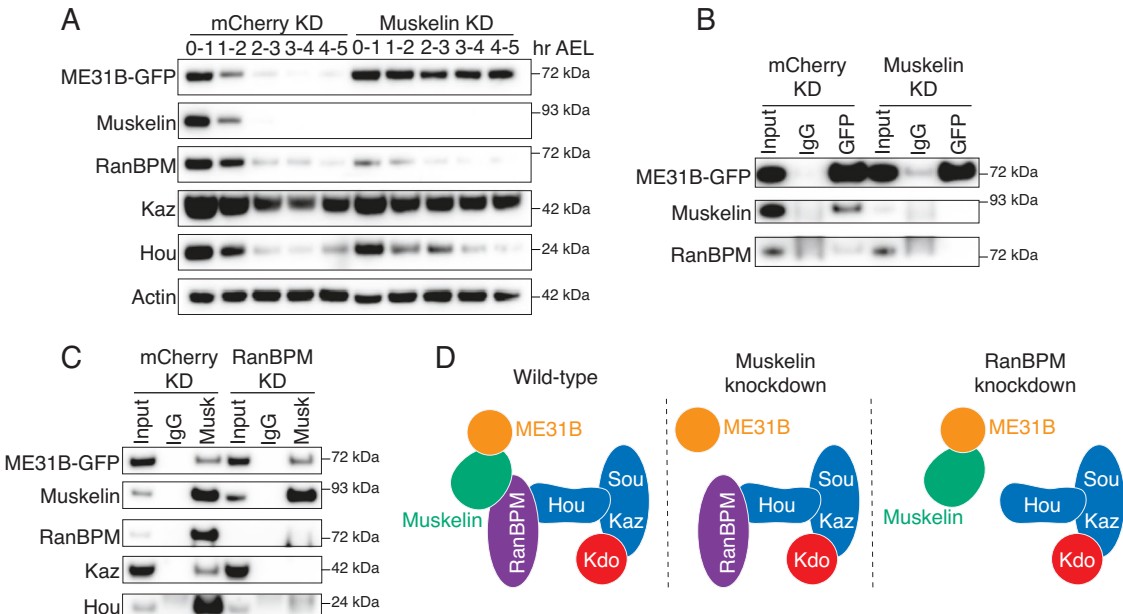

**Figure 3. Muskelin bridges the interaction between ME31B and the CTLH complex.**

(A) Muskelin depletion stabilizes ME31B-GFP. ME31B-GFP and CTLH component protein levels were probed in staged lysates from control embryos (mCherry KD) and embryos lacking Muskelin (Muskelin KD). Hr AEL: hours after egg-laying. (B) ME31B-GFP interacts with RanBPM only in the presence of Muskelin. ME31B-GFP was immunoprecipitated from 0 to 1 h control (mCherry) or Muskelin knockdown embryo lysate and probed for ME31B-GFP, Muskelin, and RanBPM. (C) Muskelin and ME31B-GFP interact independently of the CTLH complex. Muskelin was immunoprecipitated from lysates from staged 0 to 1 h control (mCherry) or RanBPM knockdown embryos, and samples were blotted for CTLH complex components and ME31B-GFP. All western blots are representative of three biological replicates. (D) Model: Muskelin bridges the interaction between ME31B and the CTLH complex via its interaction with RanBPM, and RanBPM is not required for the interaction between Muskelin and ME31B. Source data are available online for this figure.

observed in ovaries where a hypomorphic *ovo-B* allele has been rescued by OVO overexpression (Fig. 2F; Benner et al, 2024). Similarly, *Kdo*, whose transcript levels are also higher in the ovary than other adult tissues (Fig. 2A), contains OVO-binding motifs, has ChIP-seq signal at the promoter, and is transcriptionally rescued by OVO overexpression in *ovo-B* hypomorphs (Fig. EV2G). Thus, we conclude that *Muskelin* and *Kdo* are targets of OVO and expressed early in oogenesis, although in the case of *Kdo* its mRNA remains translationally repressed until egg activation (Zavortink et al, 2020; Eichhorn et al, 2016). Together, these results demonstrate that OVO is a key transcription factor for *Muskelin* expression, thus explaining its restriction to the ovary while still being broadly detectable within different cell types in ovarioles. This specific developmental control of Muskelin expression is essential for understanding its role in the embryonic CTLH complex.

## Muskelin bridges the interaction between ME31B and the CTLH complex

Given that *Muskelin* expression is developmentally restricted to the ovary, we were curious about its role within the *Drosophila* CTLH complex. Recent work on the human and yeast versions of the CTLH has shown that the WD40 β-propeller-containing components, like Muskelin, often play a structural role in the complex, allowing the generation of a massive, "chelated" E3 ligase (Sherpa et al, 2021; Qiao et al, 2020; Gottemukkala et al, 2024; Yi et al, 2024; Barbulescu et al, 2024). At the same time, in humans, WDR26 (which also contains this

same domain) has been shown to act as a substrate adaptor for the complex in lieu of the canonical Gid4 subunit (Gross et al, 2024; Maitland et al, 2024; Mohamed et al, 2021). We reasoned that Muskelin might either enable a structural conformation required for interaction with ME31B or, possibly, act as the substrate receptor itself.

We first confirmed, as we have shown previously (Zavortink et al, 2020), that degradation of ME31B-GFP depends upon Muskelin as does the interaction between ME31B and the CTLH complex in 0–1 h embryos (as monitored by RanBPM; Fig. 3A,B). In contrast, different results emerged when RanBPM was depleted (Fig. 3C). Although the interaction between Muskelin and other components of the CTLH complex (i.e., Hou and Kaz) was readily detectable in control knockdown embryos, these interactions were no longer detected in the absence of RanBPM. This result is consistent with known structures where RanBPM (and its orthologs) helps connect the WD40 β-propeller subunits to the rest of the complex (van gen Hassend et al, 2023; Sherpa et al, 2022; Lampert et al, 2018; Barbulescu et al, 2024). In contrast, Muskelin still interacted with ME31B in the absence of RanBPM. In other words, although Muskelin is required for the interaction between ME31B and the CTLH complex, RanBPM is not required for the interaction between Muskelin and ME31B (Fig. 3D).

## Muskelin is the substrate recognition adaptor of the embryonic CTLH complex

Given the lack of a known Gid4 substrate receptor homolog in flies, we explored the possibility of an unknown component of the

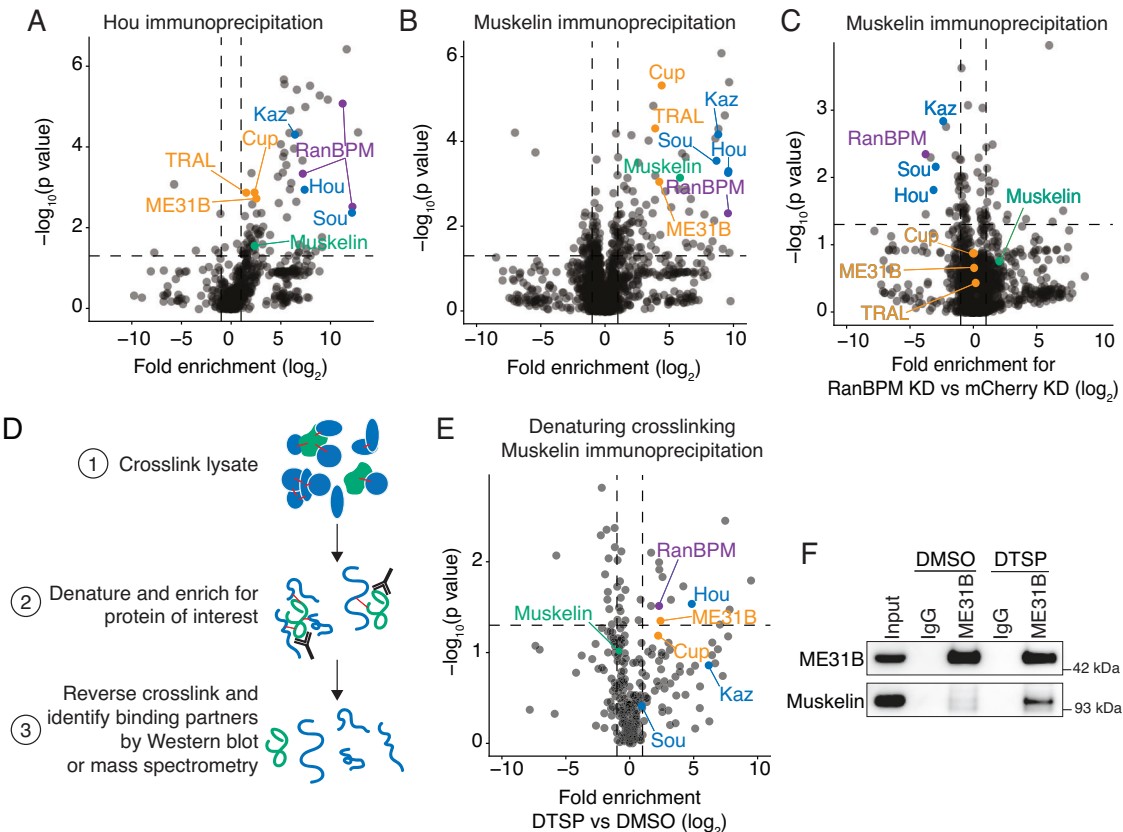

**Figure 4. Muskelin is the substrate recognition adaptor of the embryonic CTLH complex.**

(A) Hou immunoprecipitation enriches CTLH complex components and targets. Hou was immunoprecipitated from 0 to 1 h embryo lysate and binding partners were identified by mass spectrometry. Fold enrichment over control immunoprecipitation was calculated and plotted against adjusted $P$ value by $t$ test across three biological replicates. (B) Muskelin immunoprecipitation enriches CTLH complex components and targets. Muskelin was immunoprecipitated from lysates from control 0–1 h embryos and binding partners were identified by mass spectrometry. Fold enrichment over control immunoprecipitation was calculated and plotted against adjusted $P$ value by $t$ test across three biological replicates. (C) RanBPM depletion disrupts the interaction between Muskelin and the CTLH complex but not its targets. Muskelin was immunoprecipitated from 0 to 1 h control embryo lysate or embryo lysate lacking RanBPM, and binding partners were identified by mass spectrometry. The fold enrichment of each protein in RanBPM knockdown compared to control knockdown samples was calculated and plotted against adjusted $P$ value by $t$ test across three biological replicates. (D) Denaturing cross-linking immunoprecipitation workflow. Lysates are cross-linked with DTSP, denatured and immunoprecipitated, then the crosslink is reversed, and binding partners are identified. (E) Denaturing cross-linking Muskelin immunoprecipitation identifies CTLH components and targets as close-proximity binding partners in 0–1 h $w1118$ embryos by mass spectrometry. Fold enrichment of each protein from a Muskelin immunoprecipitation in DTSP-treated (cross-linked) lysate vs DMSO-treated (uncrosslinked) lysate was calculated and plotted against adjusted $P$ value by $t$ test across three biological replicates. (F) The ME31B-Muskelin interaction is detectable only in cross-linked samples from a denaturing cross-linking ME31B immunoprecipitation. ME31B was immunoprecipitated from 0 to 1 h $w1118$ embryo lysate under denaturing conditions after cross-linking with DTSP or DMSO as a control. Lysates were then probed for ME31B and Muskelin. All western blots are representative of three biological replicates. Source data are available online for this figure.

embryonic CTLH complex that acts in this role and employed a series of complementary immunoprecipitation/mass spectrometry experiments and analyses (Fig. 4A–C; Datasets EV1–3). First, we used Hou as bait to define the embryonic complex. We confirmed that Hou immunoprecipitations from 0 to 1 h embryo lysate contained RanBPM, Muskelin, Kaz, and ME31B by western blotting, indicating that our antibody was able to pull-down Hou in the context of the full complex (Fig. EV3A). Next, we performed mass spectrometry on immunoprecipitated proteins (Dataset EV1). As expected, relative to the IgG control, known targets (ME31B, Cup, and TRAL) were significantly enriched in the pulldowns as were previously identified components of the CTLH complex (Fig. 4A). YPEL5 (Yippee in flies) was identified in the Hou pulldown, which is a known homolog of a known human CTLH component that acts to inhibit substrate ubiquitylation

(Gottemukkala et al, 2024). However, this protein was not required for degradation of ME31B nor did its absence appear to substantially enhance ME31B degradation (Fig. EV3B). Although there were several other significantly enriched proteins in the Hou pulldown, none were homologs of known substrate adaptors nor had an obvious role or connection with the CTLH complex, protein degradation, or ME31B, based on annotations in FlyBase (Dataset EV1).

Second, we cross-referenced all significantly enriched proteins from published Muskelin-GFP and ME31B-GFP immunoprecipitation mass spectrometry datasets (Cao et al, 2020; Zavortink et al, 2020). Given that a putative, unknown substrate adaptor would likely interact with both Muskelin and ME31B, we reasoned it should be present in both datasets (which robustly detected the ME31B-Muskelin interaction). Once again, no homologs of known substrate adaptors were

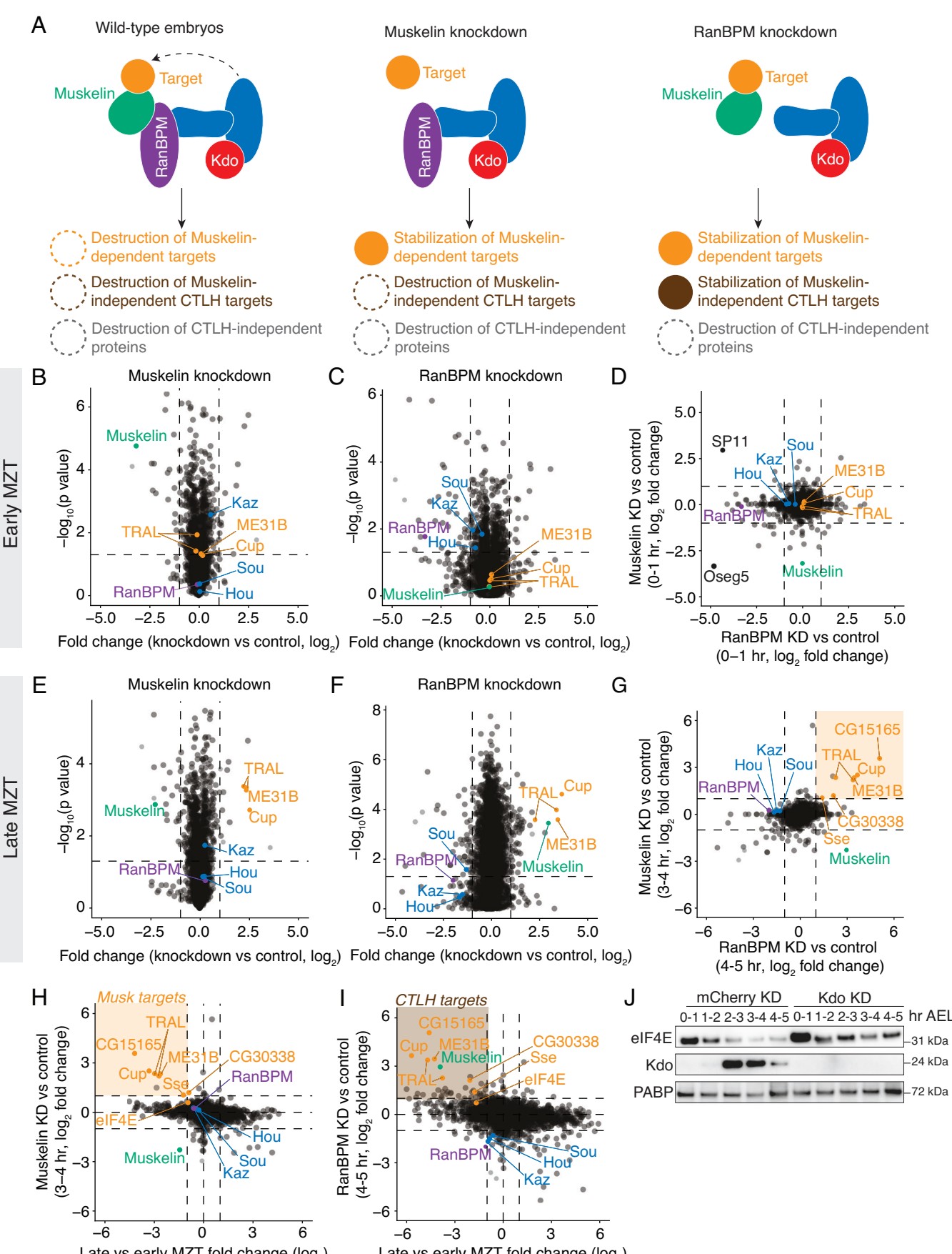

Figure 5. The main targets of the embryonic CTLH complex are ME31B, Cup, and TRAL.

(A) Muskelin regulates a distinct subset of the CTLH-dependent proteome. In wild-type embryos, many types of proteins are degraded that are both dependent on and independent of CTLH activity. In Muskelin knockdown embryos, only proteins that are specifically targeted by Muskelin are stabilized, while other CTLH targets are still degraded. In RanBPM knockdown embryos, all CTLH target proteins are stabilized, including those targeted by Muskelin. (B) Lack of Muskelin in the early embryo has minor effects on the proteome. The fold changes of protein levels in 0–1 h Muskelin knockdown embryos compared to control knockdown embryos were plotted against adjusted *P* value by *t* test. All points represent the average fold change of a specific gene across three biological replicates. (C) Lack of RanBPM in the early embryo has minor effects on the proteome. Whole-proteome fold changes of 0–1 h RanBPM knockdown embryos were compared to 0–1 h control embryos and plotted against adjusted *P* value by *t* test across three biological replicates, otherwise as in (A). (D) There are few proteomic differences between early embryos lacking RanBPM and those lacking Muskelin. Whole-proteome fold changes of 0–1 h Muskelin knockdown embryos were compared to those of 0–1 h RanBPM knockdown embryos across three biological replicates. (E) ME31B, Cup, and TRAL are significantly stabilized in late-MZT embryos lacking Muskelin. Levels of other CTLH components do not change substantially. Whole-proteome fold changes of 3–4 h Muskelin knockdown embryos were compared to 3–4 h control embryos and plotted against adjusted *P* value by *t* test across three biological replicates. (F) Muskelin, ME31B, Cup, and TRAL are significantly stabilized in late-MZT embryos lacking RanBPM. Whole-proteome fold changes of 4–5 h RanBPM knockdown embryos were compared to 4–5 h control embryos and plotted against adjusted *P* value by *t* test across three biological replicates. (G) A small subset of the proteome is stabilized in the absence of RanBPM or Muskelin (orange box). Whole-proteome fold changes of 3–4 h Muskelin knockdown embryos were compared to 4–5 h RanBPM knockdown embryos across three biological replicates. (H) Few proteins that are stabilized in embryos lacking Muskelin are also degraded in wild-type embryos during the MZT (orange box). Proteins stabilized in 3–4 h Muskelin knockdown embryos (D) were plotted against proteins degraded at the end of the MZT in control embryos across three biological replicates. (I) Few proteins that are stabilized in embryos lacking RanBPM are also degraded in wild-type embryos during the MZT (brown box). Proteins stabilized in 4–5 h RanBPM knockdown embryos (E) were plotted against proteins degraded at the end of the MZT in control embryos across three biological replicates. (J) eIF4E is mildly stabilized in embryos depleted of Kdo. eIF4E, Kdo, and PABP (as a loading control) protein levels were probed in staged lysates from control embryos and embryos lacking Kdo. Hr AEL hours after egg-laying. All western blots are representative of three biological replicates. Source data are available online for this figure.

present in the list of high-confidence enriched targets. We further cross-referenced the significantly enriched proteins between the same Muskelin-GFP immunoprecipitation dataset with our Hou immunoprecipitation mass spectrometry dataset (Fig. EV3C). None of the proteins present in both datasets were obvious homologs of CTLH components or had a role in protein degradation (Dataset EV2).

Finally, we immunoprecipitated Muskelin from 0 to 1 h embryo lysate that lacked RanBPM and performed mass spectrometry (Dataset EV3). Consistent with our previous results (Fig. 2C), in mCherry knockdown controls we were able to see significant enrichment of ME31B, Cup, and TRAL in Muskelin pulldowns relative to control pulldowns (Fig. 4B). Also consistent with previous results, in the Muskelin immunoprecipitation from embryos lacking RanBPM, the interaction with the rest of the CTLH complex was lost (Fig. 4C). Importantly, the interaction between Muskelin and ME31B, Cup, and TRAL was unchanged, supporting our model that Muskelin requires RanBPM to bridge the interaction between the CTLH complex and its targets. However, as before, no potential alternative substrate receptor was enriched.

Given these findings, we next considered the intriguing idea that Muskelin acts as the substrate receptor for the CTLH complex and explored the potential for a direct interaction between it and ME31B. To do so, we optimized denaturing cross-linking immunoprecipitations from embryo lysates (Fig. 4D). This protocol uses Dithiobis(succinimidyl propionate) (DTSP), a bivalent covalent protein crosslinker, to reversibly link all lysine residues within 12 Angstroms of one another (Akaki et al, 2022). We cross-linked native early-embryo lysate and then enriched for proteins cross-linked to Muskelin by performing immunoprecipitations under denaturing conditions with stringent washes. After reversing the crosslinks, immunoprecipitated proteins were identified by mass spectrometry in three biological replicates. When comparing the DTSP-treated condition to the DMSO-treated condition, we do not expect any changes in Muskelin enrichment because Muskelin is the bait protein for both experiments. RanBPM was significantly enriched, as was Hou (Fig. 4E; Dataset EV4). Strikingly, ME31B was also significantly enriched compared to the non-cross-linking

control (Fig. 4E; Dataset EV4). Cup was enriched but did not meet our significance cut-off of corrected *P* value < 0.05, and TRAL did not crosslink to Muskelin. Finally, eIF4E, the cap-binding protein, which interacts with Cup (Nelson et al, 2004), was also significantly and substantially enriched in the cross-linking datasets. When we performed the reverse experiment, now immunoprecipitating ME31B under denaturing conditions, we were able to detect Muskelin only in the presence of DTSP (Figs. 4F and EV3D; Dataset EV5). Together, these data are consistent with a hypothesis that Muskelin serves as the substrate receptor for the embryonic CTLH complex to target ME31B and possibly its co-repressors Cup and TRAL.

## The main degradation targets of the embryonic CTLH complex are ME31B, Cup, and TRAL

To investigate what other proteins were targeted by Muskelin and the CTLH complex, we performed total protein mass spectrometry on embryos from control, RanBPM-depleted, or Muskelin-depleted embryos in the early and late MZT (Dataset EV6). We would expect that in wild-type embryos, many proteins would be degraded regardless of their dependence on the CTLH complex (Fig. 5A). However, in Muskelin knockdown embryos, we would expect that only specific targets of Muskelin would be stabilized, while Muskelin-independent CTLH targets still degraded. Finally, in RanBPM knockdown embryos, we would expect that all CTLH targets would be stabilized, including Muskelin-dependent targets. A possible limitation of this approach is that it does not consider newly synthesized proteins, thereby potentially leading to false negatives for targets. However, there are substantial technical difficulties in labeling newly synthesized proteins in the closed system of the *Drosophila* embryo, and we focused on overall protein level changes, as these likely represent the major protein degradation targets of the E3.

Early in the MZT (at 0–1 h), there were very few changes between Muskelin knockdown and control (Fig. 5B), and only 15 proteins changed by at least twofold (with a corrected *P* value < 0.05). Consistent with a low level of Kdo at this point in embryogenesis,

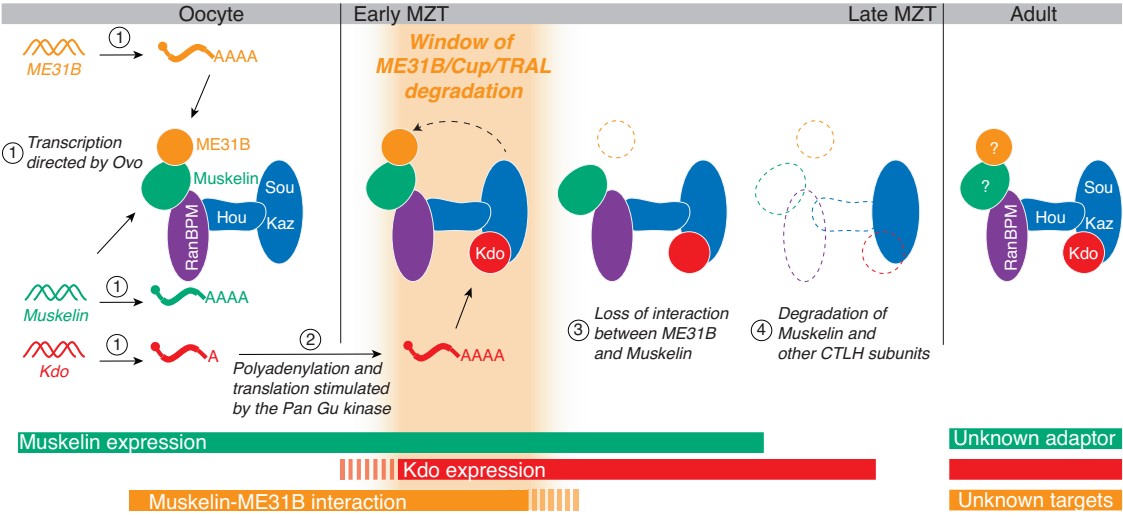

**Figure 6.  Model: At least four mechanisms restrict the degradation of ME31B and its binding partners Cup and TRAL to the MZT.**

First, in oocytes, transcription of *Muskelin, ME31B,* and *Kdo* is orchestrated by the oocyte-specific transcription factor OVO (1). *Kdo* remains translationally repressed until egg activation by activity of the Pan Gu kinase (2), but Muskelin and ME31B are translated in the oocyte. Muskelin bridges the interaction between ME31B and the CTLH complex in the developing oocyte, but because Kdo is not yet translated, the complex is "poised" and inactive. At egg activation and the beginning of the MZT, *Kdo* mRNA is translated, and its increasing protein levels activate the CTLH complex, enabling the ubiquitylation that leads to ME31B/Cup/TRAL degradation (depicted as only ME31B for clarity). At the end of the MZT, Muskelin and ME31B no longer interact (3). Finally, Muskelin, Kdo, much of the CTLH complex, and CTLH targets are degraded, effectively stopping additional degradation (4). After the MZT in the adult fly, CTLH components and Kdo are still present, but the substrate adaptor and targets in this context remain unknown. These overlapping mechanisms restrict the window of ME31B degradation to a small timeframe during the MZT.

none of the known targets of the CTLH complex (ME31B, Cup, and TRAL) were significantly different. Similar results were observed in embryos depleted of RanBPM at 0–1 h (Fig. 5C; Dataset EV6). Interestingly, several extracellular proteins (Oseg5 and SP11) were depleted in embryos lacking either Muskelin or RanBPM at this timepoint, perhaps reflecting defects that have been previously observed when Muskelin is depleted from the ovary (Kronja et al, 2014; Fig. 5D).

We next examined how loss of Muskelin or RanBPM affected the proteome at the end of the MZT (3–4 and 4–5 h after egg-laying, respectively; Fig. 5E–G). Consistent with our analysis of the autoregulation by the CTLH complex, RanBPM, Kaz, Sou, and Hou were not stabilized during the MZT in the absence of Muskelin (Fig. 5E), but Muskelin was significantly stabilized in the absence of RanBPM (Fig. 5F). Moreover, and again as expected, Cup, ME31B, and TRAL were robustly stabilized at these time points in the absence of either Muskelin or RanBPM relative to control knockdowns (Fig. 5E,F). Surprisingly, however, only 14 and 30 other proteins were upregulated in the Muskelin and RanBPM knockdown embryos, respectively (Fig. 5E,F). When we performed GO analysis on the set of significantly stabilized proteins from either dataset, no patterns emerged and no GO terms were enriched. Given that of the upregulated proteins only eight were common to both datasets (Figs. 5G and EV4), we examined their expression levels in the early embryo using published early-embryo proteomics available on FlyBase (Casas-Vila et al, 2017). All the proteins apart from Cup, TRAL, and ME31B are extremely lowly expressed, with some even being undetectable in these datasets. Similarly, within our data, the adjusted counts of these proteins are approximately twofold lower than those of Cup, TRAL, or ME31B. Still, intrigued by the possibility that these other proteins might

represent targets of the CTLH complex, we compared how levels of these proteins changed during the MZT with how they change upon loss of the E3 ligase component (Fig. 5H,I). When we plotted proteins which are degraded in wild-type embryos against proteins which are significantly stabilized in either knockdown, we noticed that three of them (CG15165, CG30338, and Sse) were both destroyed during the MZT and stabilized upon the loss of either Muskelin or RanBPM. Nonetheless, of these putative additional substrates, none were enriched in any Muskelin pulldown (Datasets EV3 and EV4), indicating that Muskelin does not serve as the substrate adaptor for these specific proteins' clearance. All three are highly helical proteins, which could potentially contribute to non-specific binding and enrichment that is not representative of Muskelin targeting them for destruction (Jochim and Arora, 2010; Edwards and Wilson, 2011; Bullock et al, 2011; Wang et al, 2021). It is also notable that there appear to be more proteins that are degraded in wild-type embryos and are stabilized in embryos lacking RanBPM than are stabilized in embryos lacking Muskelin, perhaps indicating Muskelin-independent CTLH targets.

We noticed that eIF4E, which interacts with ME31B and its binding partners and was also significantly cross-linked to Muskelin, was also slightly stabilized in the absence of RanBPM or Muskelin (Fig. 5H,I), although it failed to reach our cut-off thresholds. To investigate this observation, we probed levels of eIF4E across the MZT in control or Kdo knockdown embryos. As has been observed previously (Cao et al, 2020), eIF4E levels modestly decreased during the MZT, but were stabilized in embryos depleted of Kdo (Fig. 5J). However, the effect was much weaker than that of ME31B, Cup, and TRAL (Fig. 5H,I). Indeed, when we compare the significantly enriched proteins from late-MZT RanBPM or Muskelin knockdown embryos to the

significantly enriched proteins from a Muskelin immunoprecipitation (fold change >2 and $P$ value < 0.05), we see that only ME31B, Cup, and Tral are common to all three datasets (Fig. EV4). Taken together, these data indicate that the embryonic CTLH complex via its adaptor Muskelin regulates a very small number of proteins, and ME31B, Cup, and TRAL appear to be its major targets.

## Discussion

We have found multiple transcriptional and post-translational regulatory mechanisms that restrict the degradation of ME31B to early embryogenesis (Fig. 6). Importantly, we found that Muskelin acts as the substrate receptor for the embryonic CTLH complex. This role for Muskelin contrasts with our current understanding of the human complex, where it appears to primarily mediate multimerization of the complex rather than direct substrate recognition (van gen Hassend et al, 2023; Barbulescu et al, 2024). In the case of UNG2 in humans, Muskelin directs substrate ubiquitylation by recruitment of FAM27A (Barbulescu et al, 2024). While our cross-linking data implies that there are few other direct interactors between Muskelin and its targets, it is possible that Muskelin mediates a tripartite interaction resulting in substrate ubiquitylation without degradation of the third factor. An appealing candidate is eIF4E: it is a known cofactor for the ME31B complex, was substantially enriched in our Muskelin cross-linking immunoprecipitations, and is only weakly targeted for degradation by the CTLH complex. Similarly, in vitro, Muskelin appears to exist in an auto-tetrameric state that can shield it from binding to substrate receptors or other complex components in solution (van gen Hassend et al, 2023; Barbulescu et al, 2024). Further, while post-translational modifications (PTMs) have been detected on both human and *Drosophila* Muskelin (Hornbeck et al, 2015; Zhai et al, 2008), we have not explored the consequences of specific PTMs in the context of the MZT. Therefore, while our findings support novel and intriguing role of Muskelin as a substrate adaptor in the *Drosophila* MZT, there is much more to understand about how the CTLH complex in this context is composed on a structural level.

Like many substrate adaptors, Muskelin itself is tightly regulated, and its regulation appears to be a major mechanism restricting the degradation of ME31B to the MZT. This mechanism parallels that of other CTLH complex-mediated protein degradation, such as the autoubiquitylation of Muskelin in human cells (Maitland et al, 2019) and its regulation in response to mTOR signaling (Yi et al, 2024). This mechanism is also reminiscent of FAM72A where its high expression (and thus targeting of UNG2) only occurs in contexts where mutagenic repair levels are high (Barbulescu et al, 2024). Similarly, the expression of *Muskelin* transcripts is remarkably specific to the ovary. Our analysis points to its expression in germline stem cells, follicle cells, and the growing oocyte itself. A large part of its expression is due to OVO, a critical transcription factor in the female germline. Analysis of ChIP-seq and ATAC-seq datasets (Benner et al, 2024) indicate that the *Muskelin* promoter contains OVO-binding motifs and is indeed bound by OVO by ChIP-seq. It is interesting to note that OVO also controls *ME31B*, *Cup*, *TRAL*, and *Kdo* mRNA expression (Benner et al, 2024), demonstrating another common regulatory mechanism between the CTLH complex and its targets.

Previously, we had reported that the corresponding E2 conjugating enzyme for the CTLH complex, Kdo, was translationally controlled such that it was only produced upon egg activation (Zavortink et al, 2020). Indeed, we have now shown that the CTLH complex is able to bind its target ME31B in the mature oocyte via Muskelin, but this interaction cannot result in ubiquitylation until Kdo is produced at the onset of the MZT. Importantly, we also have now shown that Muskelin ubiquitylation, like ME31B ubiquitylation (Zavortink et al, 2020), is dependent on Kdo activity. In other words, in the oocyte, the CTLH complex appears poised to degrade its targets once translation of *Kdo* is triggered.

The CTLH complex is also post-translationally controlled. We observed that in the oocyte and early MZT, the interaction between Muskelin and ME31B is robust, but is no longer detectable by the end of the MZT. Similarly, the CTLH complex itself is degraded at the end of the MZT, and levels of nearly every member of the CTLH complex, as well as Kdo, decrease at this time in development. Given that human Muskelin is subject to autoregulation (Maitland et al, 2019) and *Drosophila* Muskelin is known to be ubiquitylated (Cao et al, 2020), we propose that the embryonic *Drosophila* CTLH complex is similarly controlled. Perhaps, this autoregulation is triggered by the loss of interaction between Muskelin and its targets to reveal sites of modification on Muskelin and/or other CTLH complex members. In support of such a mechanism, we show that the degradation of Muskelin is dependent on RanBPM and its ubiquitylation is dependent on Kdo. We also found evidence that RanBPM and Hou are co-regulated because the depletion of RanBPM led to a loss of Hou as well by western blotting (Fig. 1C). There is precedent for this finding: other work has shown that knockdown of the human ortholog of Hou (TWA1) largely destabilizes the entire CTLH complex (Lampert et al, 2018) and that TWA1 must be present in order to purify murine RanBPM (RanBP9; van gen Hassend et al, 2023).

Another surprise from our study is that, despite this complex regulatory network, the embryonic Muskelin-CTLH complex appears to have a limited number of targets, and our analysis suggests that the most important targets are Cup, ME31B, and TRAL. These three proteins are critical for oogenesis and help mediate the post-transcriptional repression of thousands of transcripts in the oocyte and embryo (Wang et al, 2017). Given that the role of the poly(A) tail and decapping change during the MZT (Wang et al, 2017), it is tempting to speculate that the degradation of ME31B-Cup-TRAL (which bind the mRNA cap-binding protein eIF4E) may be mechanistically linked to resetting the post-transcriptional landscape. Nonetheless, our finding that Muskelin knockdown does not significantly impact embryo viability (Fig. EV2F) highlights the need to understand the consequences of RNA-binding protein degradation on both maternal and zygotic mRNAs, which remains an important and unresolved issue.

Even though this work has expanded our understanding of maternal protein degradation during the MZT and the CTLH E3 ligase, it has also revealed new compelling questions. For instance, the degron recognized by Muskelin is unknown, as is whether Muskelin recognizes the ME31B-TRAL-Cup complex as a whole. That eIF4E is degraded seemingly because of its proximity to ME31B suggests that perhaps the entire complex is recognized. If so, understanding the higher-order structure of the embryonic

CTLH complex becomes more critical—how does the E3 ligase accommodate such a large protein complex, and, as a related issue, is this the mRNA-bound form? Another unanswered question is the identity or presence of other active substrate adaptors for the CTLH complex during and outside the MZT. One tempting possibility for an additional substrate adaptor is *CG7611*, the *Drosophila* ortholog of human *WDR26*. This subunit acts as the substrate adaptor in specific contexts in humans, but is not required for the degradation of ME31B and its binding partners (Gross et al, 2024; Gottemukkala et al, 2024; Zavortink et al, 2020). With recent work highlighting the dual roles of Muskelin and WDR26 (Maitland et al, 2024; Onea et al, 2022), our work provides a fascinating perspective on the evolutionary plasticity between structural components and substrate adaptors in the CTLH complex. Our results motivate further investigations into the make-up and organization of this otherwise well-conserved E3 ligase in multiple species.

## Methods

### Reagents and tools table

| Reagent/resource | Reference or source | Identifier or catalog number |
|---|---|---|
| Schnieder's S2 cells | ThermoFisher Scientific | R69007 |
| Express Five™ SFM | ThermoFisher Scientific | 10486025 |
| HiScribe T7 High Yield RNA Synthesis Kit | NEB | E2040S |
| Effectene | Qiagen | 301427 |
| TRIzol | ThermoFisher Scientific | 15596026 |
| Superscript III | ThermoFisher Scientific | 18080044 |
| iTaq SYBR Green | Bio-RAD | 1725125 |
| Phusion | ThermoFisher Scientific | F530S |
| Taq ligase | NEB | M0208L |
| T5 Exonuclease | NEB | M0663S |
| Anti-Flag ® M2 affinity gel | Millipore Sigma | A222 |
| EZView protein G affinity beads | Millipore Sigma | E3403 |
| Hoescht dye | ThermoFisher Scientific | PI62249 |
| Glass bottom imaging dishes | Ibidi | NC1437332 |
| TopTips | LC Packings | 410.992.5400 |
| TMTpro reagent | ThermoFisher Scientific | A44521 |
| Ultra-micro SpinColumn | Harvard Biosciences | 74-7206 |
| SepPak 200 mg C18 desalting cartridge | Waters | WAT054945 |
| Zorbax 2.1 mm × 150 mm Extend-C18 column | Agilent | 7737-00-902 |
| Magic $C_{18}$AQ | Michrom Bioresources | PM5-61200-00 |

| Reagent/resource | Reference or source | Identifier or catalog number |
|---|---|---|
| ReproSil-Pur $C_{18}$AQ | Dr. Maisch, Baden-Würtemburg, Germany | r13.aq.001 |
| **Experimental models** | | |
| mCherry RNAi (*D. melanogaster*) | Bloomington Drosophila Stock Center | 35785 |
| Kdo RNAi (*D. melanogaster*) | Bloomington Drosophila Stock Center | 51410 |
| RanBPM RNAi (*D. melanogaster*) | Bloomington Drosophila Stock Center | 61172 |
| Muskelin RNAi (*D. melanogaster*) | Bloomington Drosophila Stock Center | 51405 |
| **Recombinant DNA** | | |
| FI21269 | Drosophila Genomics Resource Center | |
| LD25271 | Drosophila Genomics Resource Center | |
| RH09117 | Drosophila Genomics Resource Center | |
| LD35157 | Drosophila Genomics Resource Center | |
| LD21247 | Drosophila Genomics Resource Center | |
| LP01609 | Drosophila Genomics Resource Center | |
| pIHEU-FlipGFP(Casp3 cleavage seq) T2A mCherry | Addgene | 124434 |
| **Antibodies** | | |
| Rabbit anti-Kdo | Pacific Immunology | |
| Rabbit anti-Muskelin | Boster Biological Technology | DZ41124 |
| Rabbit anti-Katazuke | Boster Biological Technology | DZ41126 |
| Rabbit anti-Houki | Boster Biological Technology | DZ41125 |
| Rabbit anti-ME31B | Boster Biological Technology | DZ33938-1 |
| Rabbit anti-RanBPM | Boster Biological Technology | DZ41192 |
| Rabbit anti-eIF4E | Boster Biological Technology | DZ41193 |
| Rabbit anti-PABP | Boster Biological Technology | DZ41256 |
| Mouse anti-GFP | Roche | 11814460001 |
| Rabbit anti-IgG | Abcam | 7074S |
| Mouse anti-Ubiquitin | Cell Signaling | 3936 |
| Mouse anti-Fas3 | DHSB | 7G10 |
| Mouse anti-Bam | DHSB | bam |
| Guinea pig anti-Tj | Oliver lab | |

| Reagent/resource | Reference or source | Identifier or catalog number |
|---|---|---|
| Donkey anti-rabbit AlexaFluor 647 | ThermoFisher Scientific | A31573 |
| Goat anti-mouse AlexaFluor 568 | ThermoFisher Scientific | A31573 |
| Goat anti-guinea pig AlexaFluor 594 | ThermoFisher Scientific | A-11076 |
| **Oligonucleotides and other sequence-based reagents** | | |
| PCR primers | This study | Table EV7 |
| Drosophila melanogaster protein database | Uniprot | UP000000803 |
| **Chemicals, enzymes, and other reagents** | | |
| DTSP | Millipore Sigma | D3669 |
| DMSO | Millipore Sigma | D2650 |
| **Software** | | |
| FIJI | | |
| LeicaX | | |
| SlideBook | | |
| Proteome Discoverer 2.5 | ThermoFisher Scientific | |
| **Other** | | |
| Line scanning confocal microscope | Stellaris | |
| Inverted spinning disk confocal microscope | Marianas | |
| Stereomicroscope | Zeiss | |
| Thermo Scientific Easy1200 nLC | ThermoFisher Scientific | LC140 |
| Orbitrap Fusion | ThermoFisher Scientific | FSN01-10002 |
| Orbitrap Eclipse with FAIMS Pro | Field Asymmetric Ion Mobility Spectrometry | FSN04-10000/ FMS02-10001 |
| Orbitrap Ascend with FAIMS Pro Duo mass spectrometer | ThermoFisher Scientific | FSN06-10000/ FMS03-1001 |

## Drosophila fly stocks

Fly stocks were maintained at 22 °C with 50% humidity. OVO fly lines were generated and maintained according to Benner et al, 2024 and maintained by the Oliver lab. All knockdowns were generated from commercially available UAS-RNAi lines crossed to a ME31B-GFP;7062 Gal4 driver (Zavortink et al, 2020). RNAi lines: mCherry (BDSC 35785), Kdo (BDSC 51410), RanBPM (BDSC 61172), Muskelin (BDSC 51405).

## S2 cells

Schnieder's S2 cells (ThermoFisher Scientific) were grown at 28 °C and maintained in media containing Express Five™ SFM (Thermo-Fisher Scientific) with 10% L-glutamine. S2 cells were tested and confirmed to be negative for Mycoplasma.

## S2 cell transfections

For RNAi, cDNA was generated by amplifying coding region of interest (obtained as plasmids from DGRC) with bidirectional T7 promoters (see Reagents and Tools Table). cDNA was transcribed using HiScribe T7 High Yield RNA Synthesis Kit (NEB) according to the manufacturer's directions. RNA was annealed slowly overnight, and dsRNA was transfected using Effectene (Qiagen) according to the manufacturer's instructions on the same day as plating. Cells were harvested and lysed in Lysis Buffer A after 72 h.

For co-transfections, S2 cells were transfected with the indicated construct using Effectene (Qiagen) 24 h after plating and harvested 48 h post-transfection.

## Cloning

For plasmids transfected in this study, inserts and backbones were amplified with gene-specific primers using CloneAmp (Takara Bio) (see Reagents and Tools Table) and Gibson Assembly Master Mix (320 μL 5× ISO buffer,

In total, 20 μL 2U/μL Phusion [ThermoFisher], 160 μL 40U/μL Taq ligase [NEB], 0.64 μL T5 Exonuclease [NEB], $H_2O$ to 1200 μL) was used to generate constructs.

## Ovary and ovariole isolation

Ovaries and ovarioles were harvested from 1 to 3-day-old virgin Drosophila according to Merkle 2023.

## Embryo and ovary lysate preparation

Embryos were collected at various time points post egg-laying on standard yeasted apple juice-agar plates at 25 °C. Embryos were dechorionated in bleach, washed in PBS, homogenized in lysis buffer A (150 mM KCl, 20 mM HEPES-KOH pH 7.4, 1 mM $MgCl_2$, 1 mM DTT, complete mini EDTA-free protease inhibitors), and were clarified at 14,000 rpm, 4 °C for 5 min. Lysates were diluted to 1 mg/mL and stored at −80 °C. Ovaries were homogenized in lysis buffer A and clarified as above.

## Antibody production

Antibodies were produced with the independent company Boster Bio using known protein sequences from UniProt, accessed via FlyBase. The antigens used for antibody production are as follows:

- Rabbit anti-Muskelin:
    MSSSSTSSPTDNSSHGSVALSSALNPSLNPVSGTNTSSQ
    KPFSLSERLNFEIYRYSSYSPNYLPENVLVDCPNDQTSRWSA
    YTNSPPQFLTLKLRRPAIVKKIKFGKFEKSHVCNIKKFRVYGG
    LDDEHMVLLLEGGLKNDNVPEVFNLRCLTEDGSENLPILYLK
    IVPLLSWGPSFNFSIWYVELHGQDDPMYTSARMKNYNSLRE-
    VEIIKLCLKHFRQQGYLSAFGALQEQTNVQLEHPLIS.
- Rabbit anti-Katazuke:
    MSEIKALEHATLKVPYEMLNKRFRSAQKIIDREVDQVINV
    SRQVEKALEAEEGPILAEVTKLVDNVAQKLQVLKRKAEESI
    NDELSVTQICKRKLDHLKGITPPSSVTGDLWQGSVDQWKRI
    RLDRLVIEHLLRMGYYETAEELAAKSDVRHLTNLDIFQTSRE

VEDDLASHSTTKCVLWCIDNKSKLRKINSTIEFSLRVQEFIE
LVRQNQRFEAVKHSRRYFPAYEKTQLNEICHVMA.

- Rabbit anti-Houki:
  MSYNEKSEAIIKEEWLQRLEQFPFKQADMNRLIMNYLVTE
  GFKEAAEKFQHEADLEPSVELSSLDGRILIREAVQAGRIEEAT
  QLVNQLHPELLGSDRYLFFHLQQLQLIELIRAGKVEEALSFAQ
  SKLSESGEEAMFELERTLALLAFEKPQESPFADLLEQSYRQKIA
  SELNSAILRCEQSEDSTPKMMFLLKLILWAQSKLDSRSISYPK
  MKNLETAHLEPK
- Rabbit anti-ME31B:
  MMTEKLNSGHTNLTSKGIINDL-
  QIAGNTSDDMGWKSKLKLPPKDNRFKTTDVTDTRGNEFE
  EFCLKRELLMGIFEKGWERPSPIQEAAIPIALSGKDVLARAK
  NGTGKTGAYCIPVLEQIDPTKDYIQALVMVPTRELALQTS
  QICIELAKHLDIRVMVTTGGTILKDDILRIYQKVQLIIATPG
  RILDLMDKKVADMSHCRILVLDEADKLLSLDFQGMLDHV
  ILKLPKDPQILLFSATFPLTV.
- Rabbit anti-RanBPM:
  QHQGVDPLRLLYPNVNESETPLPRCWSPHDKCLSIGLSQN
  NLRVTYKGVGKQHSDAASVRTAYPIPSSCGLYYFEVRIISKGR
  NGYMGIGLTAQQFRMNRLPGWDKQSYGYHGDDGNSFSSS
  GNGQTYGPTFTTGDVIGCCVNFVNNTCFYTKNGVDLGIAF
  RDLPTKLYPTVGLQTPGEEVDANFGQEPFKFDKIVDMMKE
  MRSNVLRKIDRYPHLLETPENLMNRLVSTYLVHNAFSKTAE.
- Rabbit anti-eIF4E:
  MQSDFHRMKNFANPKSMFKTSAPSTEQGRPEPPTSAAAP
  AEAKDVKPKEDPQETGEPAGNTATTTAPAGDDAVRTEHLY
  KHPLMNVWTLWYLENDRSKSWEDMQNEITSFDTVEDFWS
  LYNHIKPPSEIKLGSDYSLFKKNIRPMWEDAANKQGGRWVI
  TLNKSSKTDLDNLWLDVLLCLIGEAFDHSDQICGAVINIRG
  KSNKISIWTADGNNEEAALEIGHKLRDALRLGRNNSLQYQL
  HKDTMVKQGSNVKSIYTL.
- Rabbit anti-PABP:
  MASLYVGDLPQDVNESGLFDKFSSAGPVLSIRVCRDVIT
  RRSLGYAYVNFQQPADAERAL DTMNFDLVRNKPIRIMWSQ
  RDPSLRRSGVGNVFIKNLDRAIDNKAIYDTFSAFGNILSCK
  VATDEKGNSKGYGFVHFETEEAANTSIDKVNGMLLNGKKV
  YVGKFIPRKEREKELGEKAK LFTNVYVKNFTEDFDDEKLKEF
  FEPYGKITSYKVMSKEDGKSKGFGFVAFETTEAAEAAVQAL
  NGKDMGEGKSLYVARAQKK.

## RT-qPCR

RNA was extracted using TRIzol (ThermoFisher) according to the manufacturer's instructions. cDNA was generated using Super-Script III (ThermoFisher) according to the manufacturer's instructions and amplified with iTaq SYBR reagents (Bio-RAD) using gene-specific primers (see Reagents and Tools Table) and normalized to *Act5C*.

## Immunoprecipitations

Lysates were incubated with anti-GFP (Roche), anti-Muskelin (Boster), anti-Houki (Boster), or IgG (Abcam) antibodies and rotated for 1 h at 4 °C. EZView protein G affinity beads (Sigma) were washed with lysis buffer A 3× times. The lysate-antibody mixtures were added to 25 µL of washed protein G beads and rotated for 1 h at 4 °C. Beads were washed 3× with lysis buffer A and transferred to new tubes. Immunoprecipitations which probed

for ubiquitin were washed 6x with denaturing wash buffer (50 mM Tris pH 8, 1 M NaCl, 1% NP-40, 0.05% sodium deoxycholate, 0.1% SDS). Reducing agent (NuPAGE) and sample buffer (NuPAGE) were added to the beads prior to western blotting. Flag IPs were performed with 25 µL anti-Flag M2 beads per reaction and washed as above.

## Denaturing cross-linking immunoprecipitations

Lysates were incubated with 50 mM DTSP (Millipore Sigma) or DMSO (Millipore Sigma) for 3 h at 4 °C rotating. The reaction was quenched by adding Tris-HCl pH 7.5 to 20 mM. An equal volume 2× denaturing lysis buffer (2× PBS, 2% Triton X-100, 2% sodium deoxycholate, 2% SDS) was added followed by immunoprecipitation with anti-ME31B (Boster), anti-Muskelin (Boster), or IgG (Abcam) for 1 h at 4 °C. EZView protein G affinity beads (Sigma) were washed with lysis buffer A 3×. The lysate-antibody mixtures were added to 25 µL of washed protein G beads and rotated for 1 h at 4 °C. Beads were washed 3× with denaturing wash buffer (50 mM Tris pH 8, 1 M NaCl, 1% NP-40, 0.05% sodium deoxycholate, 0.1% SDS) and transferred to new tubes. Samples were reduced with DTT to reverse crosslink prior to western blotting or mass spectrometry analysis.

## Western blotting

Samples were run on standard Bis-Tris 4–12% gels (Invitrogen) and transferred to PVDF membrane prior to western blotting. Blots were blocked in 5% milk in PBST (PBS + 0.1% Tween) prior to incubation with primary antibodies: Rabbit anti-Kdo (Pacific Immunology) was used at 1:1000. Rabbit anti-Muskelin (Boster) was used at 1:5000. Rabbit anti-Katazuke (Boster) was used at 1:5000. Rabbit anti-Houki (Boster) was used at 1:500. Rabbit anti-ME31B (Boster) was used at 1:5000. Rabbit anti-RanBPM (Boster) was used at 1:5000. Mouse anti-GFP (Roche) was used at 1:1000. Rabbit anti-eIF4E (Boster) was used at 1:10000. Rabbit anti-PABP (Boster) was used at 1:10,000. Mouse anti-ubiquitin (Cell Signaling) was used at 1:1000.

## Immunostaining

Ovaries and ovarioles were dissected, fixed, and immunostained, according to Merkle et al, 2023. Briefly, whole ovaries were fixed in 4% paraformaldehyde in PBST (PBS + 0.1% Triton X-100) for 15 min at room temperature. Following fixation, they were blocked in 5% BSA in PBST at 4 °C overnight. Ovaries were washed with PBST, then the primary antibody was added for 4 °C overnight (rabbit anti-Muskelin 1:500 [Boster], mouse anti-Fas3 1:100 [DHSB], mouse anti-bam 1:250 [DHSB], guinea pig anti-Tj 1:2000 [gift from Oliver lab]). Following a second wash, fluorescent secondaries were added overnight at 4 °C (donkey anti-rabbit AlexaFluor 647 1:500, goat anti-mouse AlexaFluor 568 1:500, goat anti-guinea pig AlexaFluor 594 1:500; all ThermoFisher). Hoescht dye (1:1000, ThermoFisher) was added for five minutes during washes after secondary antibody incubation and ovaries were stored at 4 °C in PBST until ready for mounting. Ovarioles were hand-dissected from whole fixed ovaries prior to imaging.

Embryos were dechorionated with bleach, then fixed in a heptane/4% paraformaldehyde solution for 25 min at room

temperature. Embryos were then devitellinized with an equal volume of methanol and stored at −20 °C for at least 24 h. After rehydration with PBST (1× PBS, 0.1% Triton X-100), embryos were blocked with 10% donkey serum (Millipore Sigma) in PBST overnight at 4 °C followed by co-staining with anti-Muskelin (1:500) and anti-GFP (1:200) overnight at 4 °C. Secondaries (goat anti-mouse AlexaFluor 568 1:1000, donkey anti-rabbit AlexaFluor 647 1:1000) were incubated for 2 h at room temperature. Hoescht dye (1:1000, ThermoFisher) was added for five minutes during washes after secondary antibody incubation and embryos were stored at 4 °C in PBST until ready for mounting.

## Microscopy

Immunostained ovarioles and embryos were prepared for imaging by embedding in 0.2% agar in PBS on glass bottom imaging dishes (Ibidi). Immunostained ovariole samples were imaged with a ×10 objective on a Stellaris 8 line scanning confocal microscope and LeicaX software available from Dr. Ning Zhao's lab. Immunostained embryo samples were imaged with a ×20 objective on a 3I MARIANAS inverted spinning disk confocal microscope with SlideBook software available through the CU Advanced Light Microscopy Core. Image processing was performed in FIJI and maximum Z-projections and linear lookup tables were used for each image.

Brightfield images of ovaries and ovarioles were acquired by dissecting ovaries and preparing as above. Images were acquired with an 8X objective on a Zeiss Discovery v8 microscope with ZEN 3.4 software.

## Co-IP mass spectrometry

### On-bead digestion

Beads were washed three times with 50 mM HEPES pH 8.7 and suspended in 50 mM HEPES pH 8.7. Disulfide bonds within the proteins were reduced by the addition of tris (2-carboxyethyl) phosphine (TCEP) to a concentration of 5 mM with mixing at room temperature for 15 min. The reduced proteins were alkylated by the addition of 2-chloroacetamide to a concentration of 10 mM with mixing in the dark at room temperature for 30 min. Excess 2-chloroacetamide was quenched by the addition of dithiothreitol to a concentration of 10 mM and mixing at room temperature for 30 min. Samples were digested by the addition of 500 ng of rLys-C with mixing at 37 °C for 4 h followed by the addition of 500 ng of trypsin protease and mixing overnight at 37 °C. Samples were filtered over 2-µm spin cartridges and dried in a SpeedVac. Samples were desalted over TopTips (LC Packings), eluted with 15 mM ammonium formate pH 2.8, and dried in a SpeedVac. Samples were resuspended in 20 µL 2% acetonitrile/0.1% formic acid in preparation for LC/MS analysis.

### TMT multiplexed sample preparation

A streamlined TMT protocol was followed for multiplexed quantitative proteomics experiments (Navarrete-Perea et al, 2018). In short, initial solutions containing 100 µg of protein were in 100 µL of a buffer containing 20 mM HEPES, 100 mM KCl, 0.1 mM EDTA, 0.4% IGEPAL, 10% glycerol, 1 mM dithiothreitol, and protease inhibitor. Disulfide bonds were reduced by the addition of TCEP to a concentration of 5 mM with mixing at room

temperature for 15 min. Protein alkylation was carried out by the addition of 2-chloroacetamide to a concentration of 10 mM with mixing in the dark at room temperature for 30 min. Samples were then subjected to protein precipitation using methanol/chloroform. Precipitated protein pellets were resuspended in 70 µL of 50 mM HEPES pH 8.7 and labeled with TMTpro reagent (ThermoFisher Scientific) following the manufacturer's instructions. Upon completion of labeling, 2 µL from each sample was combined to use as a label check. The resulting pool was briefly placed in a SpeedVac to remove acetonitrile, desalted on an Ultra-micro SpinColumn (Harvard Biosciences), and the elution was taken to dryness. The pool was resuspended in 2% acetonitrile/0.1% formic acid and analyzed on an Orbitrap Fusion to measure the extent of labeling. With the labeling measured to be greater than 97% complete, each labeling reaction was quenched by the addition of hydroxylamine to a final concentration of 0.5%. Samples were combined by adding equal amounts of protein from each sample to the final pool based on total reporter ion intensity measured in the labeling check. The resulting pool was placed in a SpeedVac to remove acetonitrile followed by desalting on a SepPak 200 mg C18 desalting cartridge (Waters) and taken to dryness. The combined TMT sample was resuspended in 100 µL of 10 mM ammonium bicarbonate pH 8. The material was loaded onto a Zorbax 2.1 mm × 150 mm (5 µm particle size) Extend-C18 column (Agilent) for basic reverse-phase fractionation. Ninety-six fractions were collected and combined into 24 pools by concatenation. The pools were subsequently taken to near-dryness by vacuum centrifugation and brought up in 20 µL 2% acetonitrile/0.1% formic acid in preparation for LC/MS analysis.

## Mass spectrometry analysis

Samples from co-immunoprecipitation experiments were analyzed by LC/ESI MS/MS with a Thermo Scientific Easy1200 nLC (Thermo Scientific) coupled to an Orbitrap Fusion, Orbitrap Eclipse with FAIMS Pro (Field Asymmetric Ion Mobility Spectrometry), or Orbitrap Ascend with FAIMS Pro Duo (Orbitraps are manufactured by Thermo Scientific) mass spectrometer. In-line desalting was accomplished using a reversed-phase trap column (100 µm × 20 mm) packed with Magic $C_{18}$AQ (5-µm 200 Å resin; Michrom Bioresources, Auburn, CA) followed by peptide separations on a reversed-phase column (75 µm × 270 mm) packed with ReproSil-Pur $C_{18}$AQ (3-µm, 120 Å resin; Dr. Maisch, Baden-Würtemburg, Germany) directly mounted on the electrospray ion source. A 90-min gradient from 8% to 30% B (80% acetonitrile/0.1% formic acid) at a flow rate of 300 nL/min was used for chromatographic separations. LC/MS/MS data were collected using data-dependent acquisition methods with the MS survey scans detected in the Orbitrap. MS/MS spectra were detected in the linear ion trap using HCD activation. Selected ions were dynamically excluded for 60 s after a repeat count of 1. Data analysis was performed using Proteome Discoverer 2.5 (Thermo Scientific). The data were searched against a Uniprot *Drosophila melanogaster* (UP000000803 downloaded 3-07-21) protein database that included common contaminants (Mellacheruvu et al, 2013). Searches were performed with settings for the proteolytic enzyme trypsin. Maximum missed cleavages were set to 2. The precursor ion tolerance was set to 10 ppm and the fragment ion tolerance was set to 0.6 Da. Dynamic peptide modifications included oxidation (+ 15.995 Da on M). Dynamic modifications on the protein terminus included acetyl (+ 42.011 Da on N-terminus), Met-loss (-131.040 Da on M) and Met-loss+Acetyl (-89.030 Da on M), and static modification of

carbamidomethyl (+ 57.021 on C). Minora was used for peak abundance and retention time alignment. Sequest HT was used for database searching. All search results were run through Percolator for scoring and the false discovery rate was set to 1% at the peptide level. Raw quantitative results were transformed to $\log_2$ scale and normalized to the median value across samples. Missing data were imputed with half of the global minimum value. *P* values for pairwise comparisons were calculated by *t* test.

Multiplexed quantitative MS data were collected on a Thermo Scientific Easy1200 nLC coupled to an Orbitrap Eclipse with FAIMS Pro (Thermo Scientific) mass spectrometer. In-line desalting was accomplished using a reversed-phase trap column (100 µm × 20 mm) packed with Magic $C_{18}$AQ (5-µm, 200 Å resin; Michrom Bioresources) followed by peptide separations on a reversed-phase column (75 µm × 270 mm) packed with ReproSil-Pur $C_{18}$AQ (3-µm, 120 Å resin; Dr. Maisch, Baden-Würtemburg, Germany) directly mounted on the electrospray ion source. A 180-minute gradient from 4% to 44% B (80% acetonitrile/0.1% formic acid) at a flow rate of 300 nL/min was used for chromatographic separations. The FAIMS Pro source used varied compensation voltages of −40 V, −60 V, −80 V while the Orbitrap Eclipse instrument was operated in the data-dependent mode. MS survey scans were collected in the Orbitrap (resolution 120,000) with a 3 s cycle time, and MS/MS spectra acquisition was detected in the linear ion trap with CID activation using turbo speed scan. Selected ions were dynamically excluded for 60 s after a repeat count of 1. Following MS2 acquisition, real-time searching (RTS) was employed, and spectra were searched against a *Drosophila melanogaster* (UP000000803) protein database using the real-time search algorithm COMET to identify suitable MS/MS ions for synchronous precursor selection (SPS) MS3 quantitation. SPS-MS3 was collected on the top 10 most intense ions detected in MS2 and subjected to higher energy collision-induced dissociation (HCD) for fragmentation and subsequent analysis in the Orbitrap (resolution 50,000). Data analysis was performed using Proteome Discoverer 2.5 (Thermo Scientific). The data were searched against a *Drosophila melanogaster* (UP000000803 downloaded 03-07-21) protein database that included common contaminants (cRAPome). Searches were performed with settings for the proteolytic enzyme trypsin. Maximum missed cleavages were set to 2. The precursor ion tolerance was set to 10 ppm and the fragment ion tolerance was set to 0.6 Da. Dynamic peptide modifications included oxidation (+ 15.995 Da on M). Dynamic modifications on the protein terminus included acetyl (+ 42.011 Da on N-terminus), Met-loss (−131.040 Da on M), and Met-loss+Acetyl (−89.030 Da on M). Static modifications included TMTpro (+ 304.207 Da on any N-terminus), TMTpro (+ 304.207 Da on K), and carbamidomethyl (+ 57.021 on C). Sequest HT was used for database searching. All search results were run through Percolator for scoring and the false discovery rate was set to 1% at the peptide level. Raw quantitative results were transformed to $\log_2$ scale and normalized to the median value across samples. Missing data were imputed with half of the global minimum value. *P* values for pairwise comparisons were calculated by *t* test.

## Publicly available dataset analysis

For publicly available RNAseq and mass spectrometry data from adult fly tissues and staged embryos, data was accessed from

Flybase (release FB2024_02) (Öztürk-Çolak et al, 2024; Li et al, 2003; Rust et al, 2020; Tritschler et al, 2008).

## Experimental study design and statistics

All experiments were performed with a sample size of three biological replicates unless otherwise noted. Samples consisted of pooled biological material and randomization was not required. Blinding was not performed in any sample collection or analysis. Samples were not excluded from analysis. *T* tests were performed as statistical analysis when appropriate unless otherwise noted.

## Data availability

The mass spectrometry data from this publication have been deposited to the MassIVE database [https://massive.ucsd.edu/ProteoSAFe/static/massive.jsp] and assigned the identifiers: Mass spectrometry; Muskelin immunoprecipitation from RanBPM knockdown: MSV000094996 https://massive.ucsd.edu/ProteoSAFe/dataset.jsp?task=359a0ac56ad540678665ad52f5e8c942. Mass spectrometry; RanBPM knockdown whole proteome: MSV000094997 https://massive.ucsd.edu/ProteoSAFe/dataset.jsp?task=10325d52d6b04cfda969ffd13be34f3d. Mass spectrometry; ME31B denaturing cross-linking immunoprecipitation: MSV000094998. https://massive.ucsd.edu/ProteoSAFe/dataset.jsp?task=0d52e95095e74c02b0edce86843c99fc. Mass spectrometry; Muskelin denaturing cross-linking immunoprecipitation: MSV000094999. https://massive.ucsd.edu/ProteoSAFe/dataset.jsp?task=e145941039b944a2b38d316a07f9728c. Mass spectrometry; Hou immunoprecipitation: MSV000095000 https://massive.ucsd.edu/ProteoSAFe/dataset.jsp?task=8a2b4645969a40bc95f9f2b25ceae8eb. Mass spectrometry; Muskelin knockdown whole proteome: MSV000095001 https://massive.ucsd.edu/ProteoSAFe/dataset.jsp?task=8cf4cbea05974ac1b89ee688b98898ea. Datasets are public.

The source data of this paper are collected in the following database record: biostudies:S-SCDT-10_1038-S44319-025-00397-6.

## Peer review information

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

## Acknowledgements

The authors thank Dr. Ning Zhao and her lab for the use of their microscope; the CU Advanced Light Microscopy Core; the Oliver lab for the use of the anti-Tj antibody; C Smibert for helpful instruction in embryo collection; the Rissland lab for helpful discussion and review; and S Jagannathan, S Ramachandran, and S Nachtergaele for scientific and writing input. This work was funded by NIH grants 5T32GM136444 (CAB), 5T32GM141742 (ABR), R35GM128680 (OSR), and by NSF grant CAREER 2056136 (OSR). The Fred Hutch Proteomics and Metabolomics Shared Resource, RRID:SCR_022618, is supported by the Fred Hutch/University of Washington/Seattle Children's Cancer Consortium (P30 CA015704). The Orbitrap Ascend mass spectrometer used in this research was funded by a grant from the NIH Office of Research Infrastructure Programs (S10 OD030225). This research was supported in part by the Intramural Research Program of the NIH, The National Institute of Diabetes and Digestive and Kidney Diseases (NIDDK) (awarded to Brian Oliver). Some reagents were supplied through the Drosophila Genomics Resource Center, supported by NIH grant 2P40OD010949.

## Author contributions

**Chloe A Briney**: Data curation; Investigation; Visualization; Writing—original draft; Writing—review and editing. **Jesslyn C Henriksen**: Data curation; Investigation; Visualization; Writing—original draft; Writing—review and editing. **Chenwei Lin**: Formal analysis. **Lisa A Jones**: Formal analysis. **Leif Benner**: Data curation; Formal analysis. **Addison B Rains**: Data curation; Investigation. **Roxana Gutierrez**: Data curation; Investigation. **Philip R Gafken**: Formal analysis; Funding acquisition. **Olivia S Rissland**: Conceptualization; Supervision; Funding acquisition; Validation; Visualization; Writing—original draft; Writing—review and editing.

Source data underlying figure panels in this paper may have individual authorship assigned. Where available, figure panel/source data authorship is listed in the following database record: biostudies:S-SCDT-10_1038-S44319-025-00397-6.

## Disclosure and competing interests statement

The authors declare no competing interests.

# Expanded View Figures

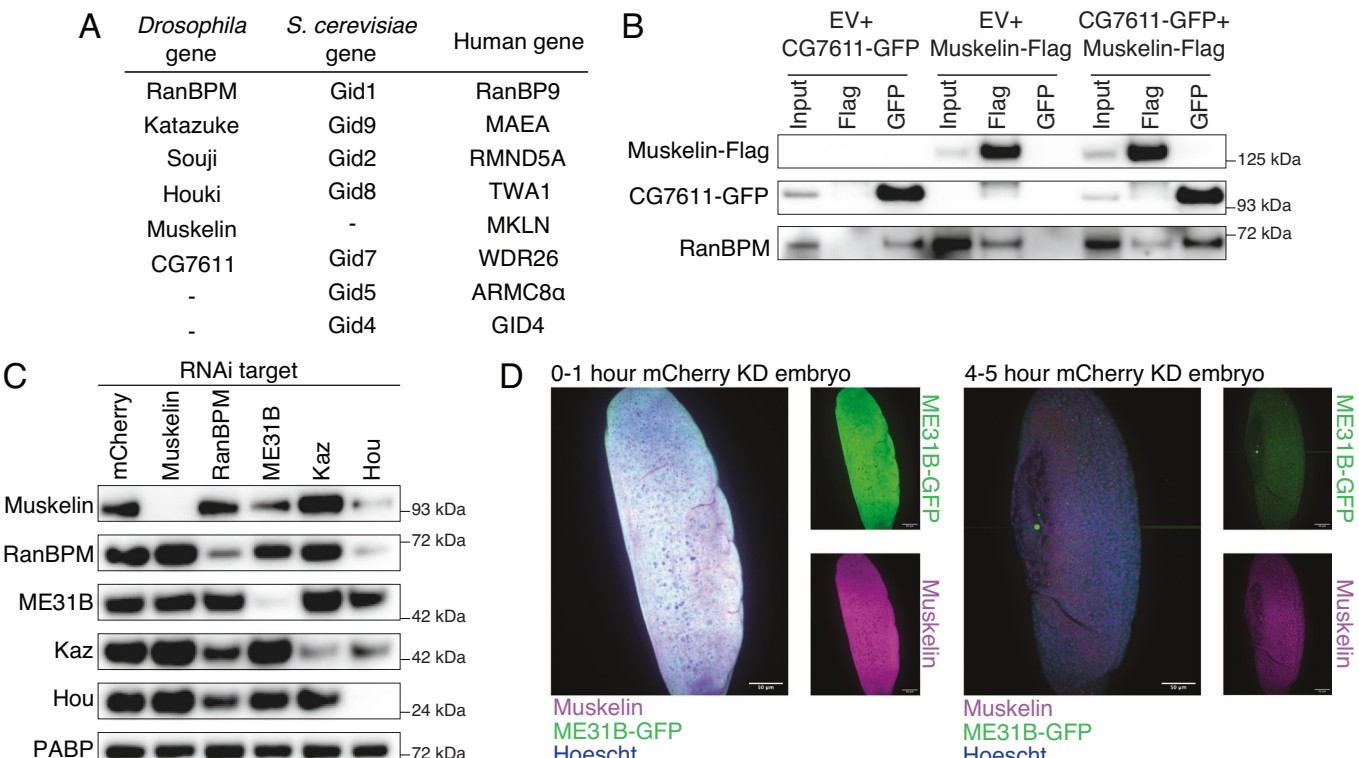

**Figure EV1. Gene name comparisons, tool validation, and co-expression analyses.**

(A) Gene name comparisons from yeast, human, and fly CTLH complex homologs. (B) CG7611 and Muskelin are mutually exclusive in their binding to RanBPM in S2 cells. CG7611-GFP and/or Muskelin-Flag were transfected into S2 cells and lysates were immunoprecipitated with anti-Flag beads or anti-GFP. (C) Antibody validation. Protein targets were depleted by RNAi in S2 cells followed by western blotting for the target using newly generated antibodies. Gene names above the blots indicate RNAi target, gene names next to blot indicate western blot target. PABP is used as a loading control. (D) Immunofluorescence for ME31B-GFP and Muskelin in embryos shows similar localization patterns to immunoprecipitations. 0–1 h and 4–5 h mCherry knockdown (control) embryos were stained with anti-GFP and anti-Muskelin. The signal is diffuse, overlapping, and strong in 0–1 h embryos and decreases noticeably in 4–5 h embryos. Scale bar = 50 µm. Western blots and microscopy images are representative images of three biological replicates.

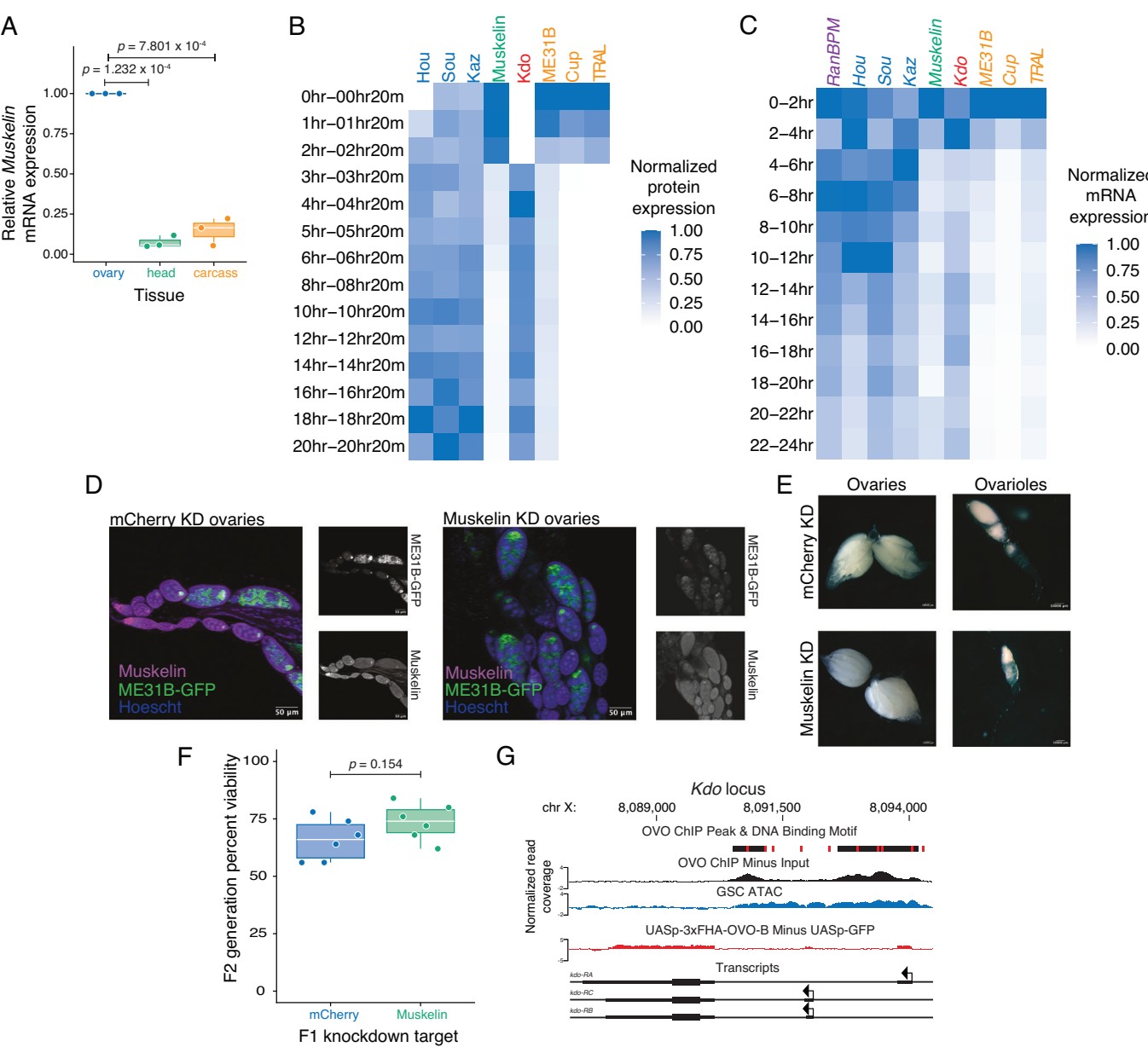

**Figure EV2. *Muskelin* expression confirmation, developmental CTLH complex component and target gene expression, immunofluorescence confirmation, Muskelin knockdown phenotypes, and *Kdo* transcriptional control.**

(A) qPCR confirmation of *Muskelin* mRNA expression in ovary, head, and carcass, normalized to *Muskelin* expression in ovary. mRNA was extracted from each tissue, converted to cDNA, and amplified using gene-specific primers. Expression levels were compared with a Student's *t* test, $n = 3$, $P = 1.232 \times 10^{-4}$ comparing ovary to head expression; $P = 7.081 \times 10^{-4}$ comparing ovary to carcass expression. Boxplot for *Muskelin* mRNA expression in the head: minimum = 0.00183, maximum = 0.0364, median = 0.00565, upper bound = 0.0210, lower bound = 0.00374, mean = 0.0146. Boxplot for *Muskelin* mRNA expression in the carcass: minimum = 0.00378, maximum = 0.0898, median = 0.0605, upper bound = 0.0752, lower bound = 0.0321, mean = 0.0514. (B, C) Embryo protein (B) and mRNA (C) expression of CTLH components and targets is highest early in development. Each gene was normalized to its maximum expression at each timepoint. (D) Muskelin immunofluorescence signal is specific to Muskelin expression. Immunofluorescence was performed on Muskelin-depleted ovaries that express ME31B-GFP. Images are representative of $n = 3$ biological replicates. Scale bar = 50 μm. This mCherry knockdown image is the same image as is used in Fig. 2D. (E) Brightfield images of mCherry knockdown or Muskelin knockdown ovaries. There are mild phenotypic differences between control and Muskelin knockdown ovaries or ovarioles. Images are representative of $n = 4$ biological replicates. Scale bar = 10,000 μm. (F) F2 viability studies for Muskelin knockdown embryos. F2 embryos from F1 mCherry or Muskelin knockdown parents do not show differences in overall viability. $N = 6$, $P = 0.154$ by Welch two-sample t test. Boxplot for mCherry knockdown F2 percent viability: minimum = 56, maximum = 78, median = 66, upper bound = 72.5, lower bound = 58, mean = 66. Boxplot for Muskelin knockdown F2 percent viability: minimum = 62, maximum = 84, median = 74, upper bound = 79, lower bound = 69, mean = 73.67. (G) *Kdo* gene level read coverage tracks for OVO ChIP minus input, GSC ATAC-seq, and *ovo^{ΔBP}/ovo^{ovo-GAL4}; UASp-3xFHA-OVO-B* minus *ovo^{ΔBP}/ovo^{ovo-GAL4}; UASp-GFP* RNAseq. Red and black rectangles represent significant OVO DNA binding motifs and OVO ChIP peaks, respectively. Gene models are represented at bottom. Small rectangles represent untranslated regions, large rectangles represent translated regions. Arrows indicate transcriptional start sites.

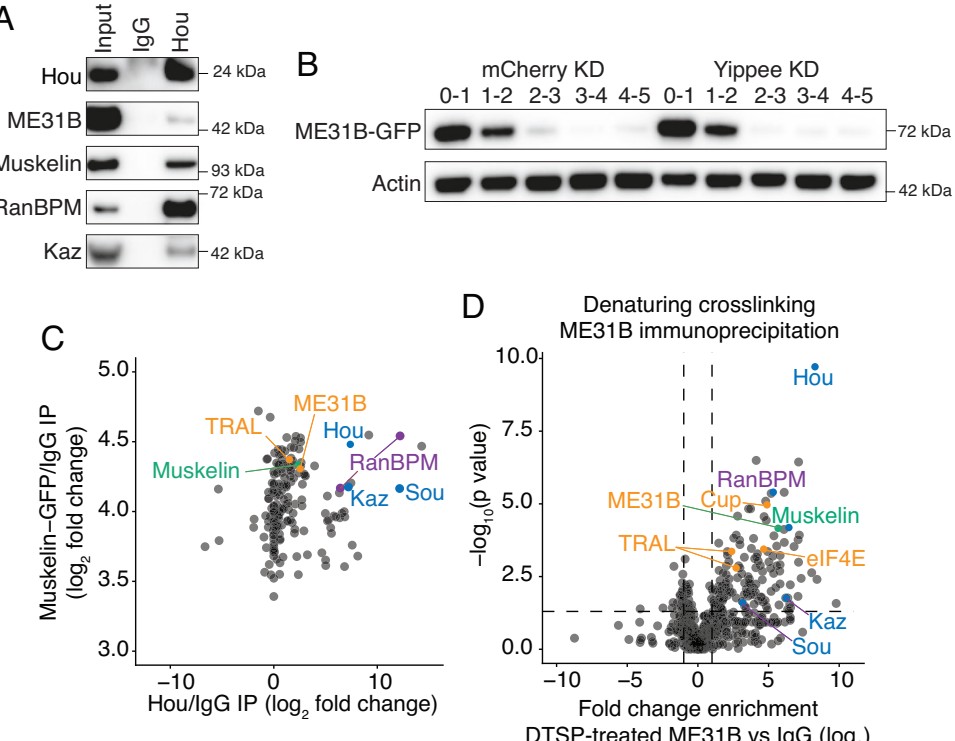

**Figure EV3. Hou immunoprecipitation, Yippee knockdown, and denaturing crosslinking ME31B immunoprecipitation mass spectrometry.**

(A) Hou immunoprecipitation with the newly developed antibody enriches for CTLH components. Hou was immunoprecipitated from *w1118* 1–2 h embryo lysate and probed for CTLH complex components. (B) Yippee knockdown does not stabilize ME31B-GFP. Embryo lysates were collected across the MZT time course from control embryos or embryos lacking Yippee and probed for ME31B-GFP and actin (as a loading control). (C) No putative substrate adaptors are identifiable from comparing Hou immunoprecipitations to Muskelin-GFP immunoprecipitations. Hou immunoprecipitation fold changes compared to control immunoprecipitation were compared to Muskelin-GFP over control fold changes from Cao et al, 2020. In all plots, blue points represent CTLH complex components, orange points represent targets, green points represent bait protein. Each point represents the average fold change of a specific gene across *n* = 3 biological replicates. (D) Denaturing cross-linking ME31B immunoprecipitation mass spectrometry comparing cross-linked ME31B immunoprecipitation to cross-linked control immunoprecipitation demonstrates CTLH component and repressor complex components as close-proximity binding partners of ME31B in 0–1 h *w1118* embryo lysate. Fold changes of ME31B IP in lysates treated with DTSP were compared to IgG IP in lysates treated with DTSP and plotted against the *P* value by *t* test. All western blots are representative of three biological replicates.

A

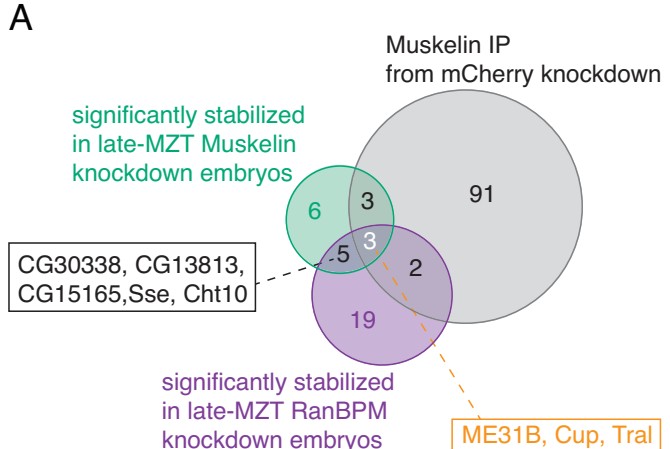

Muskelin IP
from mCherry knockdown

significantly stabilized
in late-MZT Muskelin
knockdown embryos

6 3 91

CG30338, CG13813,
CG15165,Sse, Cht10

-5 2

significantly stabilized
in late-MZT RanBPM
knockdown embryos

ME31B, Cup, Tral

**Figure EV4. Venn diagram comparing significantly stabilized proteins from knockdown experiments with Muskelin interactors from immunoprecipitation.**

(A) Venn diagram comparing significantly stabilized proteins (fold change >2, *P* value, 0.05) from Muskelin knockdown embryos, RanBPM knockdown embryos, and Muskelin immunoprecipitation from mCherry knockdown embryos. Only ME31B, Cup, and Tral are common to all three datasets, highlighting Muskelin's target specificity. Data from this study (Datasets EV3 and EV6).

