## [Peer Review File · EMBO Reports]

Muskelin is a substrate receptor of the highly-regulated *Drosophila* embryonic CTLH E3 ligase

Chloe Briney, Jesslyn Henriksen, Chenwei Lin, Lisa Jones, Leif Benner, Addison Rains, Roxana Gutierrez, Philip Gafken, and Olivia Rissland

Corresponding author(s): Olivia Rissland (olivia.rissland@cuanschutz.edu)

Review Timeline:

Submission Date:	24th Jul 24
Editorial Decision:	29th Aug 24
Revision Received:	3rd Jan 25
Editorial Decision:	29th Jan 25
Revision Received:	31st Jan 25
Accepted:	5th Feb 25

Editor: Esther Schnapp

Transaction Report:

Dear Olivia,

Thank you for the transfer of your manuscript to EMBO reports. We have now received the full set of referee reports that is pasted below.

As you will see, the referees acknowledge that the findings are potentially interesting. However, they also have several suggestions for how the study could or should be strengthened. All raised points are reasonable, and I think that potentially all should be addressed, but please let me know what you think and we can discuss the exact revision requirements further, also in a video chat, if you like. The best way forward may be that you send me a proposed revision plan that we can discuss and that I can potentially also discuss with the referees, if necessary.

I would thus like to invite you to revise your manuscript with the understanding that the referee concerns must be fully addressed and their suggestions taken on board. Please address all referee concerns in a complete point-by-point response. Acceptance of the manuscript will depend on a positive outcome of a second round of review. It is EMBO reports policy to allow a single round of major revision only and acceptance or rejection of the manuscript will therefore depend on the completeness of your responses included in the next, final version of the manuscript.

We realize that it is difficult to revise to a specific deadline. In the interest of protecting the conceptual advance provided by the work, we recommend a revision within 3 months (29th Nov 2024). Please discuss the revision progress ahead of this time with the editor if you require more time to complete the revisions.

- 1) A data availability section providing access to data deposited in public databases is missing. If you have not deposited any data, please add a sentence to the data availability section that explains that.
- 2) Your manuscript contains statistics and error bars based on $n=2$. Please use scatter blots in these cases. No statistics should be calculated if $n=2$.

3) We replaced Supplementary Information with Expanded View (EV) Figures and Tables that are collapsible/expandable online. A maximum of 5 EV Figures can be typeset. EV Figures should be cited as 'Figure EV1, Figure EV2' etc... in the text and their respective legends should be included in the main text after the legends of regular figures.

5) a complete author checklist, which you can download from our author guidelines <https://www.embopress.org/page/journal/14693178/authorguide>. Please insert information in the checklist that is also reflected in the manuscript. The completed author checklist will also be part of the RPF.

6) Please note that all corresponding authors are required to supply an ORCID ID for their name upon submission of a revised manuscript (<https://orcid.org/>). Please find instructions on how to link your ORCID ID to your account in our manuscript tracking system in our Author guidelines

<<https://www.embopress.org/page/journal/14693178/authorguide#authorshippinguidelines>>

12) All Materials and Methods need to be described in the main text using our 'Structured Methods' format, which is required for all research articles. According to this format, the Methods section includes a Reagents and Tools Table (listing key reagents, experimental models, software and relevant equipment and including their sources and relevant identifiers) followed by a Methods and Protocols section describing the methods using a step-by-step protocol format. The aim is to facilitate adoption of the methodologies across labs. More information on how to adhere to this format as well as a downloadable template (.docx) for the Reagents and Tools Table can be found in our author guidelines:

An example of a Method paper with Structured Methods can be found here: <https://www.embopress.org/doi/full/10.1038/s44320-024-00037-6#sec-4>

As part of the EMBO publication's Transparent Editorial Process, EMBO reports publishes online a Review Process File (RPF) to accompany accepted manuscripts. This File will be published in conjunction with your paper and will include the referee

reports, your point-by-point response and all pertinent correspondence relating to the manuscript.

I look forward to seeing a revised form of your manuscript when it is ready.

Best wishes,
Esther

Referee #1:

In this paper, the authors follow up on their previous findings that ME31B, Cup, and Trailer Hitch (TRAL) are ubiquitinated by the CTLH E3 ligase complex and Kdo in *Drosophila*. These proteins are implicated in the maternal-to-zygotic transition during embryogenesis, where the CTLH targets the RNA-binding protein ME31B and its two partners, Cup and TRAL. However, the regulation of this process has been unclear. They identify several layers of regulation, including Ovo-mediated transcriptional control of Muskulin during oogenesis and specific binding of the three proteins to the Muskulin subunit of the CTLH. The manuscript is clearly written overall, and the mechanisms characterized and proposed in the current study provide interesting insights into the CTLH E3 complex in flies, with some similarities and disparities with the yeast GID and human CTLH complex. I only have a few minor comments that could potentially clarify some aspects of the authors' description.

1. Related to Fig. 1. "It is tempting to speculate the reduced interaction may in fact trigger the autoregulation of the CTLH complex."
This model is hard to grasp as ME31B consistently binds to MKLN1 even in Kdo KO cells. For this model to be valid, one would imagine that the MKLN1-ME31B interaction may be lost due to some modifications on either side of the protein, regardless of Kdo activity. If the authors intended to convey that ME31B degradation triggers free MKLN1 formation, then it would be expected that ME31B and MKLN1 form a steady state stoichiometric complex, but this seems not the case based on the overall co-IP efficiency. Additionally, based on the blot in Fig. 1C, it appears that MKLN1 is degraded faster than ME31B within 1-2 hours. Therefore, restating this model would be beneficial for readers to comprehend what exactly the authors are proposing.
 2. Related to Figure 2F: Would mutating the predicted OVO binding sites in the Muskulin or Kdo promoters prevent transcriptional regulation? While the authors' hypothesis is logically understandable, more experimental data could give it credibility, if possible.
 3. Related to Fig. 4D and E, this does not seem to provide additional information to the findings in Fig. 4 A-C. Could the authors possibly map the cross-linked sites?
 4. Figure 5: In performing total protein mass spectrometry, the authors are trying to emphasize protein degradation but do not account for newly synthesized proteins.
 5. Figure 4-5. Overall, the authors focus on the known interactors of the baits or expected hits. Although many proteins are enriched more than the expected hits in various proteomics datasets, they are not labeled or discussed. What are those statistically significant hits beyond the CTLH E3 complex? For example, in Fig. 5B,C, and especially E, there are strongly enriched/depleted proteins after KD, but they are not labeled. Perhaps labeling them and commenting on those specific proteins would be informative, as they are the results of unbiased experiments..
 6. Overall, adding detailed definitions or annotations for each column in the supplemental tables will greatly assist the readers. The current use of abbreviations is making it difficult for readers to comprehend the data.
 7. "Muskulin is mutually exclusive with WDR26 and intriguingly, it is both ubiquitinated as part of autoregulation by the CTLH complex and regulates a distinct subset of the proteome compared to WDR26."
- In the Maitland et al. 2024 paper, Muskulin is "proposed" to be mutually exclusive with WDR26. However, it should be noted that this work does not rule out the alternative possibility that the two adaptors could coexist. Therefore, it might be potentially more accurate to include the word "proposed." here.
8. Third, transcription of the Muskulin subunit (which is required for ME31B degradation) is tightly restricted to oogenesis.

'Third' seems typo. It should change to Second.

9. Method section: Precipitated protein pellets were resuspended in 70 μ L of 50 mM HEPES pH 8.7 and labeled with TMTpro reagent.

It looks like there may be a typo in "HEPES" - it might be "EPPS".

Referee #2:

This manuscript focuses on an important biological process - the maternal-to-zygotic transition (MZT) - in a genetic model which has pioneered mechanistic understanding of the process (*Drosophila*). Two previous papers published in 2020 - Zavortink et al. and Cao et al. - had shown that the CTLH E3 ligase complex targets the posttranscriptional repressive complex comprised of Cup, TRAL and ME 31B (referred to here as C-T-M) for clearance by the proteasome mid-MZT. The current study focuses on the CTLH complex itself, most notably the Muskelin subunit, and provides evidence, of particular note and quite surprisingly, that Muskelin is likely to be carrying out substrate recognition of C-T-M. Evidence is also provided for another surprising conclusion - that apart from C-T-M, there appear to be few additional targets of Muskelin/CTLH during the MZT.

In general, the manuscript is well-written, (most of) the data clearly presented, and (most of) the conclusions warranted. I do, however, have several concerns that need to be addressed. (In the absence of line-numbering, I will refer to the page and paragraph numbers.)

Major:

- 1) P. 5, par. 4 - P. 6, par. 1; also P. 13, par. 2: Production of antibodies and western blots. The description of antibody design, production, and testing is inadequate. Where are the controls for specificity of the new antibodies (westerns and IF)?
- 2) P. 5, par. 4 - P. 6, par. 1: Casas Vila et al. 2017 (on FlyBase) present high resolution mass spec data for Souji, Katazuke, Houki and Muskelin. The Casas Vila data for Muskelin are consistent with that shown in Cao et al. 2020 and the current study by western blots. However, the Casas Vila data for Souji (stable), Katazuke (stable) and Houki (up at 1 hr then stable) are different from the western blots in the current study. Do the authors have an explanation for this?
- 3) P. 6, par. 2: The data presented are consistent with the possibility of autoregulation/autoubiquitylation of *Drosophila* Muskelin but do not directly address whether it is (auto)ubiquitylated. It would bolster their argument if the authors mentioned that it has previously been shown that *Drosophila* Muskelin is ubiquitylated during the MZT (Cao et al. 2020, Table S3).
- 4) P. 6, par. 2 & 3: I find the data in Fig. 2C & D inadequate in several aspects. First, there are no controls to show that the antibody staining for Muskelin is specific rather than background; second, the magnification and resolution are inadequate; third, because of this the claim for "colocalization as puncta" is not supported either for colocalization of for puncta. See also (5) below.
- 5) P. 6-8: I find the entire section on "Muskelin expression is restricted to oogenesis" to be a digression and distraction from the main topic of the manuscript. Furthermore, as described in (4) the data are unconvincing. Those data appear to show expression of ME31B in both the germline and soma of the ovary, the latter aspect of which is not understood in terms of function. Furthermore, the fact that Muskelin appears to be a transcriptional target of OVO is not surprising in light of the large number of germline genes regulated by OVO. To focus the manuscript on the most interesting and novel aspects of the study I recommend deletion of this section.
- 6) P. 8-10: This section on "Muskelin is the substrate recognition adaptor of the embryonic CTLH complex" is the most novel and interesting part of the manuscript. I find most of the data convincing but am a little confused by some aspects of the figures. Fig. 4B: Why is TRAL shown twice, once significantly enriched, once not? Fig. 4C and text: The authors need to be careful how they describe the data. The only significantly enriched proteins are in the top right quadrant. When they state that interaction with ME31B, Cup, and TRAL is "maintained" I presume that the point is that these are not depleted. Rewriting is required to clarify this. Fig. 4E: The authors should clarify in the text that Muskelin is not expected to be enriched given that the x-axis is the ratio of DTSP to DMSO. Where is TRAL?
- 7) Fig. 5 H & I: The axes of this figure are complicated and not well explained in the text. Are the authors sure that the x-axis ratio is late/early not early/late? In other words, please explain why Muskelin and CTLH targets are in the top left rather than the top right quadrant.

Minor:

- 1) P.3, par. 1 and Reference list P. 20: Cao et al. 2021 should be Cao et al. 2022

2) P. 4, par. 3: The authors should add that *Drosophila* Muskelin was shown previously to be cleared during the late MZT (Cao et al. 2020 Fig. 6B).

3) P. 5, par. 2: "First....third...". Where is "Second"?

4) P. 5, par. 3:

"ME31B degradation in the MZT is mediated by the E2-E3 combination of Marie Kondo (Kdo) and the CTLH complex (Zavortink et al. 2020; Cao et al. 2020), but little is known about its degradation beyond the MZT. To investigate the temporal control of ME31B degradation, we performed an extended time course by harvesting protein lysate from embryos through and beyond the end of the MZT."

This is a bit of an overstatement. Cao et al. 2020 Fig. 2A presented a high-resolution western blot time course from 0-6 hr. The current study presents a lower resolution time course from 0-8 hr. (Fig. 1 B).

It would be more accurate to state: "ME31B degradation in the MZT is mediated by the E2-E3 combination of Marie Kondo (Kdo) and the CTLH complex (Zavortink et al. 2020; Cao et al. 2020). A previous study examined degradation of ME31B, Cup, TRAL and other proteins through and beyond the MZT (Cao et al. 2020). To further investigate the temporal control of ME31B degradation, we performed an extended time course by harvesting protein lysate from embryos to 8 hr."

5) P. 7, par. 1:

"...Muskelin is required for the degradation of ME31B, Cup, and TRAL (Zavortink et al. 2020)..."

Revise to "...Muskelin is required for the degradation of ME31B, Cup, and TRAL (Zavortink et al. 2020, Cao et al. 2020)..."

Referee #3:

The CTLH E3 ligase, a multi-subunit RING-type ligase, plays conserved role in regulating various biological processes. In contrast to human and other species, the *Drosophila* genome lacks orthologs for the characterized substrate adaptor. In this work, the authors provide evidence showing that Muskelin is a substrate adaptor for the *Drosophila* CTLH complex, and that Muskelin has exquisite target specificity for ME31B and its cofactors. Generally, the work is of potential interest in the field. However, the reviewer has a number of questions that need to be addressed.

1. In Fig. 1C, the expression levels of Muskelin in Kdo KD embryos are almost consistent within the 0-5h period. However, in Fig. 1D, the levels of Muskelin in the input samples at 0-1h, 2-3h, and 4-5h in Kdo KD are significantly different (the expression level at 0-1h is significantly higher than that at 2-3h and 4-5h). How can this be explained? Additionally, we believe that the statement "suggesting that the decrease in levels of these proteins might partly lead to the reduced interaction" is not very accurate. In 4-5h embryos, Muskelin is almost undetectable, and since Muskelin is not present in the input, it is naturally impossible to detect any interaction. This conclusion seems meaningless. Given that the authors believe that the interaction between Muskelin and ME31B significantly decreases as the embryo develops and MZT progresses, in addition to CoIP, we think it might be more meaningful and accurate to perform a dynamic analysis of the co-localization of the two proteins at different developmental stages of early embryos.

2. Fig. 2C, this study found that Muskelin is specifically highly expressed in germline stem cells and follicle cells. In Fig. S2D, immunofluorescence staining of Muskelin KD ovary was also performed, but it did not explore whether the knockdown or loss of Muskelin affects the differentiation of germline stem cells and germline development. Based on images from the Fig. S2D, Muskelin KD led to aberrant germline development.

3. Are all the CHIP-seq and RNA-seq data related to OVO in Fig. 2F and Fig. S2E entirely sourced from Benner et al. 2024, or did this study also independently validate some of these data? (Based on the article's description, it seems that all the data were sourced from Benner et al. 2024). We only see the description in the methods section (*Drosophila* fly stocks) stating, "OVO fly lines were generated and maintained according to Benner et al. 2024 and maintained by the Oliver lab." Since the study mentions using OVO-related *Drosophila*, but it is not clearly stated what experiments were conducted using these OVO-related flies, this needs to be explicitly clarified.

4. The work demonstrates through denaturing crosslinking immunoprecipitations that Muskelin and ME31B may have a direct interaction (Fig. 4E and 4F), which is one of the most important conclusions of this study. To further confirm this result, it is recommended to perform *in vitro* direct interaction experiments (Pull-down and SPR assays) to validate this conclusion."

5. "Fig. 5B, 5C, 5E, 5F should indicate the developmental stages of the embryos corresponding to the samples in the figure."

6. "What impact does the knockdown or deletion of Muskelin have on embryonic development, and are there any related phenotypic data?"

We thank the reviewers for their thoughtful and close reading of the manuscript. We appreciate that the reviewers generally found our manuscript “well-written” and as “providing interesting insights” into the CTLH E3 ligase. We also agree with the consensus that the most important finding of our work is the role of Muskelin as a substrate adaptor and its “exquisite specificity” for these three RNA binding proteins. The comments and questions raised by the reviewers have strengthened our manuscript, and we hope the reviewers find it appropriate for *EMBO Reports*.

In the manuscript text, we have indicated modified text in blue. In this document, the reviewer comments are in black, and our response is in blue.

Reviewer 1

Related to Fig. 1. "It is tempting to speculate the reduced interaction may in fact trigger the autoregulation of the CTLH complex." This model is hard to grasp as ME31B consistently binds to MKLN1 even in Kdo KO cells. For this model to be valid, one would imagine that the MKLN1-ME31B interaction may be lost due to some modifications on either side of the protein, regardless of Kdo activity. If the authors intended to convey that ME31B degradation triggers free MKLN1 formation, then it would be expected that ME31B and MKLN1 form a steady state stoichiometric complex, but this seems not the case based on the overall co-IP efficiency. Additionally, based on the blot in Fig. 1C, it appears that MKLN1 is degraded faster than ME31B within 1-2 hours. Therefore, restating this model would be beneficial for readers to comprehend what exactly the authors are proposing.

The text was clarified (lines 185-187).

Related to Figure 2F: Would mutating the predicted OVO binding sites in the Muskelin or Kdo promoters prevent transcriptional regulation? While the authors' hypothesis is logically understandable, more experimental data could give it credibility, if possible.

We agree with the expectation that removing OVO binding sites should result in a loss of Muskelin and Kdo transcriptional regulation, and we saw two potential methods for testing this hypothesis. The first involved depleting OVO from the germline, but when we attempted to characterize the impact of depleting OVO on Muskelin levels, the results were largely uninterpretable. This result is likely due to the loss of OVO causing germline death and otherwise impairing oogenesis. A second option, as the reviewer suggests, is to mutate the OVO binding sites in the Muskelin or Kdo promoters. While we are interested in pursuing this question more in-depth in the future, unfortunately, generating the genetic reagents to test this option is beyond the scope of the manuscript. However, the ChIP binding patterns, motif presence, and transcriptional response to OVO activity by Muskelin and Kdo is identical to what has been observed for other known OVO targets (Bielinska et al., 2005; Lü and Oliver, 2001), strongly suggesting that OVO is responsible for the same transcriptional regulation.

Related to Fig. 4D and E, this does not seem to provide additional information to the findings in Fig. 4 A-C. Could the authors possibly map the cross-linked sites?

We agree with the reviewer that mapping cross-linking sites would be a powerful addition to our study. Unfortunately, due to the use of reversible crosslinkers, we cannot map the cross-linked

sites with our current datasets; confidently mapping (irreversible) crosslinking sites by mass-spectrometry requires substantially more material than we can obtain from embryos.

Figure 5: In performing total protein mass spectrometry, the authors are trying to emphasize protein degradation but do not account for newly synthesized proteins.

As the reviewer notes, with whole proteomics in the early embryo, it is not possible to label newly synthesized proteins, especially because amino acid stores are deposited into the egg (and thus will also be included in maternally synthesized proteins). Similar problems have also plagued the quantification of maternal RNA degradation and have required very sophisticated methods to resolve (and, in fact, are still open area of inquiry). However, given that we are focused on an E3 ligase, we feel that comparing degraded proteins – or proteins that specifically increase upon loss of the E3 ligase – is a logical approach, although it may result in seemingly false negatives for targets. We clarified the text to make this limitation clear (lines 325-329).

Figure 4-5. Overall, the authors focus on the known interactors of the baits or expected hits. Although many proteins are enriched more than the expected hits in various proteomics datasets, they are not labeled or discussed. What are those statistically significant hits beyond the CTLH E3 complex? For example, in Fig. 5B,C, and especially E, there are strongly enriched/deriched proteins after KD, but they are not labeled. Perhaps labeling them and commenting on those specific proteins would be informative, as they are the results of unbiased experiments.

We performed GO analysis of the additional proteins, and no patterns emerged for any specific targets of Muskelin or the CTLH complex. Additionally, when we compare the overlap in our datasets from Muskelin immunoprecipitations from wild-type embryos, Muskelin immunoprecipitations from RanBPM KD embryos, and stabilized targets in 4-5 hour Muskelin KD embryos, only ME31B, Cup, and TRAL emerge as shared between the datasets. We modified the text to include these analyses and provided additional discussion of the other enriched proteins from Figure 5 (lines 345-352; Figure EV4, lines 359-362, lines 371-374).

Overall, adding detailed definitions or annotations for each column in the supplemental tables will greatly assist the readers. The current use of abbreviations is making it difficult for readers to comprehend the data.

We apologize for the lack of clarity in our initial submission. The supplemental tables have been edited.

"Muskelin is mutually exclusive with WDR26 and intriguingly, it is both ubiquitylated as part of autoregulation by the CTLH complex and regulates a distinct subset of the proteome compared to WDR26." In the Maitland et al. 2024 paper, Muskelin is "proposed" to be mutually exclusive with WDR26. However, it should be noted that this work does not rule out the alternative possibility that the two adaptors could coexist. Therefore, it might be potentially more accurate to include the word "proposed." here.

We thank the reviewer for raising the point, and it prompted us to conduct additional experiments in S2 cells. Here, using tagged, ectopically expressed Muskelin or *Drosophila* WDR26 (currently known as CG7611), we found that these are mutually exclusive in their binding to RanBPM. These data strongly argue that the two adaptors exist in mutually exclusive complexes. We clarified the text to reflect these new experiments (lines 114-120, Figure EV1B).

Third, transcription of the Muskelin subunit (which is required for ME31B degradation) is tightly restricted to oogenesis.' Third' seems typo. It should change to Second.

The text has been corrected.

Method section: Precipitated protein pellets were resuspended in 70 μ L of 50 mM HEPES pH 8.7 and labeled with TMTpro reagent. It looks like there may be a typo in "HEPES" - it might be "EPPS".

The text has been corrected.

Reviewer 2

Major comments

P. 5, par. 4 - P. 6, par. 1; also P. 13, par. 2: Production of antibodies and western blots. The description of antibody design, production, and testing is inadequate. Where are the controls for specificity of the new antibodies (westerns and IF)?

We produced the antibodies with an independent company (Boster), and these details have been included in Materials and Methods. We tested specificity with a variety of westerns and immunofluorescence, as appropriate, that were included in Figures EV1 and EV2 and described in lines 154-162.

P. 5, par. 4 - P. 6, par. 1: Casas Vila et al. 2017 (on FlyBase) present high resolution mass spec data for Souji, Katazuke, Houki and Muskelin. The Casas Vila data for Muskelin are consistent with that shown in Cao et al. 2020 and the current study by western blots. However, the Casas Vila data for Souji (stable), Katazuke (stable) and Houki (up at 1 hr then stable) are different from the western blots in the current study. Do the authors have an explanation for this?

The reviewer points out a difference from the Casas Vila data that we have also noted. Interestingly, even within the Casas Vila data, there are conflicting results – for instance, with lower resolution, Houki appears to decrease between 0–2 hr and 4–6 hr (similar to our results), but is undetectable at 0–1 in the more finely resolved datasets from that same study. As we did not generate the Casas Vila datasets, it is impossible for us to know the source of their variation although it may reflect subtle differences in embryo staging; however, we have repeatedly seen the same results by mass spectrometry and western blotting.

P. 6, par. 2: The data presented are consistent with the possibility of autoregulation/autoubiquitylation of Drosophila Muskelin but do not directly address whether it is (auto)ubiquitylated. It would bolster their argument if the authors mentioned that it has previously been shown that Drosophila Muskelin is ubiquitylated during the MZT (Cao et al.2020, Table S3).

We appreciate this suggestion, and we also favor a model of autoubiquitylation. As additional support, we performed additional experiments to show that endogenous Muskelin is ubiquitylated in a Kdo-dependent manner, which are now included (Figure 1D, lines 169-174).

P. 6, par. 2 & 3: I find the data in Fig. 2C & D inadequate in several aspects. First, there are no controls to show that the antibody staining for Muskelin is specific rather than background; second, the magnification and resolution are inadequate; third, because of this the claim for "colocalization as puncta" is not supported either for colocalization of for puncta. See also (5) below.

We apologize for the confusion about the controls we were performed for showing specificity. In our previous and current version, we assayed specificity in our staining by performing it in Muskelin knock-down ovaries, where Muskelin is undetectable by western blotting (the images included in Figure EV2). In addition, we have omitted the description of "puncta" from the text as we agree that the resolution does not support the use of this terminology.

P. 6-8: I find the entire section on "Muskelin expression is restricted to oogenesis" to be a digression and distraction from the main topic of the manuscript. Furthermore, as described in (4) the data are unconvincing. Those data appear to show expression of ME31B in both the germline and soma of the ovary, the latter aspect of which is not understood in terms of function.

Furthermore, the fact that Muskelin appears to be a transcriptional target of OVO is not surprising in light of the large number of germline genes regulated by OVO. To focus the manuscript on the most interesting and novel aspects of the study I recommend deletion of this section.

We respectfully disagree with the reviewer that the finding that Muskelin expression is restricted to oogenesis is a digression. In fact, it is key for partly explaining why ME31B is not targeted outside the MZT. That Muskelin is a transcriptional target of OVO is, as the reviewer notes, unsurprising (and gratifying), but that does not change the importance of the finding for understanding the CTLH complex and its regulation during the MZT.

P. 8-10: This section on "Muskelin is the substrate recognition adaptor of the embryonic CTLH complex" is the most novel and interesting part of the manuscript. I find most of the data convincing but am a little confused by some aspects of the figures. Fig. 4B: Why is TRAL shown twice, once significantly enriched, once not? Fig. 4C and text: The authors need to be careful how they describe the data. The only significantly enriched proteins are in the top right quadrant. When they state that interaction with ME31B, Cup, and TRAL is "maintained" I presume that the point is that these are not depleted. Rewriting is required to clarify this. Fig. 4E: The authors should clarify in the text that Muskelin is not expected to be enriched given that the x-axis is the ratio of DTSP to DMSO. Where is TRAL?

We appreciate the reviewers excitement over this section, as it was the most exciting for us as well. In terms of the specific concerns raised, the double plotting of TRAL reflects the presence of different isoforms. For clarity, the other isoform has been removed from being annotated in the figure. In our description of Figure 4C, we have clarified the text in light of the reviewer's comments (lines 297-299). We have also clarified the text regarding Figure 4E (lines 308-310, line 313). Interestingly, we did not observe TRAL as specifically enriched in the crosslinking; there are both technical and biological reasons for this, related to where TRAL (and crosslink-able side chains) are in the complex, and so we have focused our attention on ME31B and Cup.

Fig. 5 H & I: The axes of this figure are complicated and not well explained in the text. Are the authors sure that the x-axis ratio is late/early not early/late? In other words, please explain why Muskelin and CTLH targets are in the top left rather than the top right quadrant.

We appreciate this feedback and apologize for the mislabeling of the axes. We have simplified and clarified the text and figure legend explanations for Figures 5H and I.

Reviewer 3

In Fig. 1C, the expression levels of Muskelin in Kdo KD embryos are almost consistent within the 0-5h period. However, in Fig. 1D, the levels of Muskelin in the input samples at 0-1h, 2-3h, and 4-5h in Kdo KD are significantly different (the expression level at 0-1h is significantly higher than that at 2-3h and 4-5h). How can this be explained? Additionally, we believe that the statement "suggesting that the decrease in levels of these proteins might partly lead to the reduced interaction" is not very accurate. In 4-5h embryos, Muskelin is almost undetectable, and since Muskelin is not present in the input, it is naturally impossible to detect any interaction. This conclusion seems meaningless. Given that the authors believe that the interaction between Muskelin and ME31B significantly decreases as the embryo develops and MZT progresses, in addition to CoIP, we think it might be more meaningful and accurate to perform a dynamic analysis of the co-localization of the two proteins at different developmental stages of early embryos.

We appreciate the thoughtful feedback from the reviewer about this result. In response to the suggestion of examining Muskelin and ME31B co-localization, we performed additional immunofluorescence experiments (Figure EV1D, lines 179-181). Both proteins show broad expression, which is consistent with their molecular interaction. In terms of the immunoprecipitations, we appreciate the concern about examining interactions when both proteins are substantially less expressed. Nonetheless, we can enrich both in their respective immunoprecipitations, and still fail to detect an interaction. Of course, given that E3-target interactions are notably unstable (in fact, that between the CTLH complex and ME31B is an outlier in that respect), we cannot formally exclude that an interaction still occurs *in vivo* and believe that our wording reflects a cautious interpretation on this front.

Fig. 2C, this study found that Muskelin is specifically highly expressed in germline stem cells and follicle cells. In Fig. S2D, immunofluorescence staining of Muskelin KD ovary was also performed, but it did not explore whether the knockdown or loss of Muskelin affects the differentiation of germline stem cells and germline development. Based on images from the Fig.2SD, Muskelin KD led to aberrant germline development.

A previous report from the Orr-Weaver lab (Kronja et al., 2014) had hinted that Muskelin depletion affects oogenesis, and we do observe subtle proteomic differences in the early embryo upon Muskelin knockdown. Nonetheless, prompted by the reviewer's comments, we examined Muskelin knockdown ovaries more closely and observed only mild phenotypic differences (Figure EV2E, lines 214-218).

Are all the CHIP-seq and RNA-seq data related to OVO in Fig. 2F and Fig. S2E entirely sourced from Benner et al. 2024, or did this study also independently validate some of these data? (Based on the article's description, it seems that all the data were sourced from Benner et al. 2024). We only see the description in the methods section (Drosophila fly stocks) stating, "OVO fly lines were generated and maintained according to Benner et al.2024 and maintained by the Oliver lab." Since the study mentions using OVO-related Drosophila, but it is not clearly stated what experiments were conducted using these OVO-related flies, this needs to be explicitly clarified.

We apologize for the confusion with our previous wording. All ChIP- and RNA-seq data are from the Benner et al 2024 study, and we have clarified the text (lines 231-232).

The work demonstrates through denaturing crosslinking immunoprecipitations that Muskelin and ME31B may have a direct interaction (Fig. 4E and 4F), which is one of the most important

conclusions of this study. To further confirm this result, it is recommended to perform *in vitro* direct interaction experiments (Pull-down and SPR assays) to validate this conclusion.

We appreciate the reviewer noting that this finding as one of the most important of our study, and we agree that *in vitro* direct interaction experiments would provide strong evidence. We also thank the reviewer for this comment, as it prompted us to propose a different molecular model.

To address this question, we have collaborated with Brenda Schulman's lab. After substantial optimization of tags on the RNA binding proteins (especially Cup), they were able to purify the Cup, ME31B, and TRAL, and test for interactions with Muskelin. However, they were not able to reproducibly detect a stable interaction (data available upon request). We can imagine several possibilities for this result, ranging from missing post-translational modifications to a missing "component."

These results prompted us to re-examine our crosslinking mass spectrometry dataset, where we noted that eIF4E shows very strong enrichment in Muskelin pull-downs upon crosslinking, in fact substantially stronger than either ME31B or Cup. Although substantially more biochemical and structural work need to be done, we now consider it very possible that eIF4E could bridge the Cup-TRAL-ME31B complex to Muskelin, in a way that is analogous to FAM72a and UNG2 (Barbulescu et al., 2024). Exploring these ideas, unfortunately, is outside the scope of our current manuscript, but we have included this potential idea in the Discussion (lines 387-399).

Fig. 5B, 5C, 5E, 5F should indicate the developmental stages of the embryos corresponding to the samples in the figure.

We have added notes in the text clarifying what developmental stages the embryos are from (line 333, line 342).

What impact does the knockdown or deletion of Muskelin have on embryonic development, and are there any related phenotypic data?

We thank the reviewer for raising this question. We have now included viability studies on embryos from Muskelin knockdown mothers (Figure EV2F). The resultant flies are viable, as might be expected given the relatively small changes in the proteome upon loss of Muskelin. Although exploring *why* the stabilization of ME31b and its cofactors has so little impact (which should essentially give a gain-of-function phenotype) is beyond the scope of our current study, we have included a brief speculation about why these embryos may not be developmentally impaired (lines 217-220).

References

- Barbulescu, P., Chana, C.K., Wong, M.K., Ben Makhlof, I., Bruce, J.P., Feng, Y., Keszei, A.F.A., Wong, C., Mohamad-Ramshan, R., McGary, L.C., Kashem, M.A., Ceccarelli, D.F., Orlicky, S., Fang, Y., Kuang, H., Mazhab-Jafari, M., Pezo, R.C., Bhagwat, A.S., Pugh, T.J., Gingras, A.-C., Sicheri, F., Martin, A., 2024. FAM72A degrades UNG2 through the GID/CTLH complex to promote mutagenic repair during antibody maturation. *Nat Commun* 15, 7541. <https://doi.org/10.1038/s41467-024-52009-x>
- Bielinska, B., Lü, J., Sturgill, D., Oliver, B., 2005. Core Promoter Sequences Contribute to ovo-B Regulation in the *Drosophila melanogaster* Germline. *Genetics* 169, 161–172. <https://doi.org/10.1534/genetics.104.033118>
- Kronja, I., Whitfield, Z.J., Yuan, B., Dzek, K., Kirkpatrick, J., Krijgsveld, J., Orr-Weaver, T.L., 2014. Quantitative proteomics reveals the dynamics of protein changes during *Drosophila* oocyte maturation and the oocyte-to-embryo transition. *Proc Natl Acad Sci USA* 111, 16023–16028. <https://doi.org/10.1073/pnas.1418657111>
- Lü, J., Oliver, B., 2001. *Drosophila* OVO regulates ovarian tumor transcription by binding unusually near the transcription start site. *Development* 128, 1671–1686. <https://doi.org/10.1242/dev.128.9.1671>

Dear Olivia,

Thank you for the submission of your revised manuscript. We have now received the enclosed reports from the referees as well as cross-comments from referee 2. Referees 2 and 3 still have a few more suggestions that I would like you to address and incorporate (along the lines suggested by referee 2) before we can proceed with the official acceptance of your manuscript.

A few editorial requests will also need to be addressed:

- Please correct the conflict of interest subheading to "Disclosure and Competing Interests Statement" and move it to after the Acknowledgments.
- There is an author name discrepancy - Leif Benner in the ms vs. Lief Benner in our online submission system, please correct.
- Please remove the author credits from the ms file. All credits need to be entered during online ms submission.
- The REFERENCE format is not correct: et al needs to be used after 10 author names; DOIs should only be used for preprints and datasets that have not been published yet. Please use the EMBO reports style.
- In the author checklist, please answer all questions on the statistics and send us a new checklist.
- This FUNDING INFO is missing in our online submission system: NIH Office of Research Infrastructure Programs (S10 OD030225); Intramural Research Program of the NIH; The National Institute of Diabetes and Digestive and Kidney Diseases (NIDDK); NIH grant 2P40OD010949
- You have uploaded 7 EV Tables, and all their legends must be removed from the ms file. Most of the EV Tables should be called Datasets (especially Table EV6); EV Tables should roughly occupy 1 page that can be properly converted to PDF with the legend on the same page, while Datasets have several columns and long rows. I would like to suggest that you rename most EV Tables to Datasets. Table EV7 could be part of the Reagents Table.
Table EV2 has "Table S2" in the sheet name - this needs to be corrected.
If you convert EV Tables to Datasets then the correct nomenclature should be Dataset EV1-EV6 and the table files as well as the ms callouts need all to be corrected.
- Materials and Methods should be just Methods
- Our systematic image analysis of ms to be accepted detected 2 issues:

Similar highlights in Figure 1C WESTERNBLOTS Muskelin / Ranbpm. The source data shows some differences but can you please explain this similarity?

There might be a cell reuse between Figure 2D and Figure EV2D that is Not listed in the figure legend. Can you please clarify ?

- Figure legends:

1. Please note that the box plots need to be defined in terms of minima, maxima, centre, bounds of box and whiskers, and percentile in the legends of figures EV2 A, F.
2. Please note that information related to n is missing in the legends of figures 5H, I.

I would like to suggest to may be change the last sentence of the abstract to :

Thus, multiple levels of integrated regulation restrict the activity of the embryonic CTLH complex to early embryogenesis, when/where it regulates three important RNA binding proteins.

EMBO press papers are accompanied online by A) a short (1-2 sentences) summary of the findings and their significance, B) 2-3 bullet points highlighting key results and C) a synopsis image that is exactly 550 pixels wide and 200-600 pixels high (the height is variable). The synopsis image should provide a sketch of the major findings, like a graphical abstract. Please note that text needs to be readable at the final size. Please send us this information along with the final manuscript.

Best wishes,
Esther

Referee #1:

The authors have done an excellent job addressing most of the reviewers' points, including my own, through new experiments and data or by revising the manuscript.

Referee #2:

The manuscript is significantly improved. There are a few remaining issues that must be addressed.

1) Production of antibodies. It is not sufficient to state that "Antibodies were produced with independent company Boster Bio using known protein sequences from UniProt, accessed via FlyBase." The authors must provide information on the sequence(s) of the antigen(s) that the company used as antigens for each protein.

2) I thank the authors for removing "puncta" but I remain concerned that the magnification and resolution in Figs. 2C and D are inadequate to claim colocalization. I recommend rewriting (line 229) as follows (revision in CAPS): "Indeed, the two proteins are CO-EXPRESSED in the developing oocyte (Figure 2D, Figure EV2D)."

There is no point-by-point response to the 'Minor' points that I listed in my original review and as far as I can tell the authors have not carried out the following requested revisions (I have removed the page and paragraph numbers used in my original review and have attempted to find the lines in the revision and then have largely kept the wording of my earlier review):

3) Line 62 and Line 747: Cao et al. 2021 should be Cao et al. 2022

4) Lines 149-155:

"ME31B degradation in the MZT is mediated by the E2-E3 combination of Marie Kondo (Kdo) and the CTLH complex (Zavortink et al. 2020; Cao et al. 2020), but little is known about its degradation beyond the MZT. To investigate the temporal control of ME31B degradation, we performed an extended time course by harvesting protein lysate from embryos through and beyond the end of the MZT."

This is a bit of an overstatement. Cao et al. 2020 Fig. 2A presented a high-resolution western blot time course from 0-6 hr which extended several hours beyond the MZT. The current study presents a lower resolution time course from 0-8 hr (Fig. 1 B).

It would be more accurate to state: "ME31B degradation in the MZT is mediated by the E2-E3 combination of Marie Kondo (Kdo) and the CTLH complex (Zavortink et al. 2020; Cao et al. 2020). A previous study examined degradation of ME31B, Cup, TRAL and other proteins through and beyond the MZT (Cao et al. 2020). To further investigate the temporal control of ME31B degradation, we performed an extended time course by harvesting protein lysate from embryos to 8 hr."

5) Line 204-205:

"...Muskelin is required for the degradation of ME31B, Cup, and TRAL (Zavortink et al. 2020)..."

Revise to

"...Muskelin is required for the degradation of ME31B, Cup, and TRAL (Zavortink et al. 2020, Cao et al. 2020)..."

Referee #3:

The revised version provides additional supportive data; however, some issues still remain. Not all statements in the paper are adequately substantiated.

The authors did not provide a reasonable explanation for the following concern: "In Fig. 1C, the expression levels of Muskelin in Kdo KD embryos are almost consistent within the 0-5h period. However, in Fig. 1D, the levels of Muskelin in the input samples at 0-1h, 2-3h, and 4-5h in Kdo KD are significantly different (the expression level at 0-1h is significantly higher than at 2-3h and 4-5h). How can this discrepancy be explained?"

The reviewer is still concerned about the conclusion that Muskelin and ME31B may have a direct interaction. Since the pull-down assay failed to demonstrate this interaction, the authors should either employ other more effective methods to prove this

conclusion or revise the conclusion section (Line 325) to ensure its accuracy, rather than merely discussing it. This study found that Muskelin acts as a substrate adaptor of the highly regulated *Drosophila* embryonic CTLH E3 ligase in early embryos. Given that knocking down Muskelin has no apparent effect on embryo development, its function in early embryos remains a concern and requires further investigation.

Cross-comments from referee 2 on referee 3's report:

The authors did not provide a reasonable explanation for the following concern: "In Fig. 1C, the expression levels of Muskelin in Kdo KD embryos are almost consistent within the 0-5h period. However, in Fig. 1D, the levels of Muskelin in the input samples at 0-1h, 2-3h, and 4-5h in Kdo KD are significantly different (the expression level at 0-1h is significantly higher than at 2-3h and 4-5h). How can this discrepancy be explained?"

I think that the reviewer is referring to new 1C and 1E. The point of the experiment was to ask whether in the absence of Kdo, Muskelin is stabilized (1C) and the Muskelin-ME31B interaction is maintained (1E). To my mind this is clear in both 1C and 1E. After mCherry KD Muskelin is down at 2-3 and almost gone/no interaction at 4-5; in contrast in both 1C and 1E after Kdo KD Muskelin levels do not decrease/interaction maintained at 4-5.

The reviewer still concerned about the conclusion that Muskelin and ME31B may have a direct interaction. Since the pull-down assay failed to demonstrate this interaction, the authors should either employ other more effective methods to prove this conclusion or revise the conclusion section (Line 325) to ensure its accuracy, rather than merely discussing it.

From my perspective further experiments would be outside the scope of the current study. The authors could perhaps tone down the last sentence of the relevant section of the Results (Line 324-326) as "Together these data are consistent with the hypothesis that Muskelin serves as the *Drosophila* CTLH's substrate receptor for ME31B and, possibly, other components of the repressive complex".

This study found that Muskelin acts as a substrate adaptor of the highly regulated *Drosophila* embryonic CTLH E3 ligase in early embryos. Given that knocking down Muskelin has no apparent effect on embryo development, its function in early embryos remains a concern and requires further investigation.

This is unfair. The current study is at the molecular/biochemical not the developmental level. Effects on embryonic development are definitely outside the scope.

Thank you for your feedback on our revised manuscript. We appreciate the formatting and clarification edits that you suggest and have responded to the reviewers' additional comments. The editorial requests have also been addressed and are briefly detailed below. New changes to the manuscript are in purple text.

Editorial comments

Smaller editorial comments which have been addressed in the manuscript or submission system:

- Disclosure and Competing Interests Statement has been corrected
- Methods section header has been corrected
- Leif Benner's name has been corrected
- Author credits have been removed from the manuscript
- Reference format has been corrected and updated
- Funding information has been included in the submission system
- Summary and key results are now included on the abstract page of the manuscript
- EV Tables 1-7 have been renamed as EV Datasets and callouts in the manuscript have been updated to reflect this.

Similar highlights in Figure 1C WESTERNBLOTS Muskelin / Ranbpm. The source data shows some differences but can you please explain this similarity?

These two blots came from the same Western blot membrane, as did the Actin and Kdo blots, and we have placed the marker images alongside the HRP images are below for clarity. The odd band shape in all four blots seems to have come from an irregularity in the gel. The blots including the marker image has also been added to the source data file for Figure 1C and uploaded to the submission system.

There might be a cell reuse between Figure 2D and Figure EV2D that is Not listed in the figure legend. Can you please clarify?

We appreciate the opportunity to clarify and note that in the legend for Figure 2D, we state that the ovarioles come from mCherry knockdown control flies. We have added additional text in the legends of Figure 2D and Figure EV2D to clarify that this is the same mCherry knockdown ovariole image.

Please note that the box plots need to be defined in terms of minima, maxima, centre, bounds of box and whiskers, and percentile in the legends of figures EV2 A, F.
These data have been added to the figure legends.

Please note that information related to n is missing in the legends of figures 5H, I.
This information has been updated in the figure legend.

In the author checklist, please answer all questions on the statistics and send us a new checklist.

The author checklist has been updated and a section has been added to the Methods detailing experimental study design and statistics (lines 726-730).

I would like to suggest to may be change the last sentence of the abstract to:
"Thus, multiple levels of integrated regulation restrict the activity of the embryonic CTLH complex to early embryogenesis, when/where it regulates three important RNA binding proteins."

We appreciate your suggestion and have changed the last sentence to something very similar:
"Thus, multiple levels of integrated regulation restrict the activity of the embryonic CTLH complex to early embryogenesis, during which time it regulates three important RNA binding proteins."

Referee #1

The authors have done an excellent job addressing most of the reviewers' points, including my own, through new experiments and data or by revising the manuscript.

In response to Referee #3: I could see both the new information this paper brings into the field, especially in terms of their relevance in development, and reviewer 3's fair criticism; I feel that it is totally the editors' decision on whether this paper brings conceptual advance, despite the missing biochemical mechanism, that aligns with the scope of EMBO reports.

We appreciate the kind feedback and agree that this manuscript brings conceptual advancements to the field within the scope of EMBO Reports.

Referee #2

The manuscript is significantly improved. There are a few remaining issues that must be addressed.

1) Production of antibodies. It is not sufficient to state that "Antibodies were produced with independent company Boster Bio using known protein sequences from UniProt, accessed via FlyBase." The authors must provide information on the sequence(s) of the antigen(s) that the company used as antigens for each protein.

We have added the sequence of each antigen used to the Methods section.

2) I thank the authors for removing "puncta" but I remain concerned that the magnification and resolution in Figs. 2C and D are inadequate to claim colocalization. I recommend rewriting (line 229) as follows (revision in CAPS): "Indeed, the two proteins are CO-EXPRESSED in the developing oocyte (Figure 2D, Figure EV2D)."

We appreciate the concern raised by the reviewer, and we have adopted the suggested wording change (line 242).

There is no point-by-response to the 'Minor' points that I listed in my original review and as far as I can tell the authors have not carried out the following requested revisions (I have removed the page and paragraph numbers used in my original review and have attempted to find the lines in the revision and then have largely kept the wording of my earlier review):

3) Line 62 and Line 747: Cao et al. 2021 should be Cao et al. 2022
We have fixed this citation.

4) Lines 149-155: "ME31B degradation in the MZT is mediated by the E2-E3 combination of Marie Kondo (Kdo) and the CTLH complex (Zavortink et al. 2020; Cao et al. 2020), but little is known about its degradation beyond the MZT. To investigate the temporal control of ME31B degradation, we performed an extended time course by harvesting protein lysate from embryos through and beyond the end of the MZT."

This is a bit of an overstatement. Cao et al. 2020 Fig. 2A presented a high-resolution western blot time course from 0-6 hr which extended several hours beyond the MZT. The current study presents a lower resolution time course from 0-8 hr (Fig. 1 B). It would be more accurate to state: "ME31B degradation in the MZT is mediated by the E2-E3 combination of Marie Kondo (Kdo) and the CTLH complex (Zavortink et al. 2020; Cao et al. 2020). A previous study examined degradation of ME31B, Cup, TRAL and other proteins through and beyond the MZT (Cao et al. 2020). To further investigate the temporal control of ME31B degradation, we performed an extended time course by harvesting protein lysate from embryos to 8 hr."

We have incorporated this edit to more accurately reflect the contributions of the Cao et al. 2020 paper and include our time course data (line 160-162).

5) Line 204-205: "...Muskelin is required for the degradation of ME31B, Cup, and TRAL (Zavortink et al. 2020)..." Revise to "...Muskelin is required for the degradation of ME31B, Cup, and TRAL (Zavortink et al. 2020, Cao et al. 2020)..."

We have updated these citations.

Referee #3

The revised version provides additional supportive data; however, some issues still remain. Not all statements in the paper are adequately substantiated.

The authors did not provide a reasonable explanation for the following concern: "In Fig. 1C, the expression levels of Muskelin in Kdo KD embryos are almost consistent within the 0-5h period. However, in Fig. 1D, the levels of Muskelin in the input samples at 0-1h, 2-3h, and 4-5h in Kdo KD are significantly different (the expression level at 0-1h is significantly higher than at 2-3h and 4-5h). How can this discrepancy be explained?"

Referee #2 response: I think that the reviewer is referring to new 1C and 1E. The point of the experiment was to ask whether in the absence of Kdo, Muskelin is stabilized (1C) and the Muskelin-ME31B interaction is maintained (1E). To my mind this is clear in both 1C and 1E. After mCherry KD Muskelin is down at 2-3 and almost gone/no interaction at 4-5; in contrast in both 1C and 1E after Kdo KD Muskelin levels do not decrease/interaction maintained at 4-5.

Referee #1 response: Regarding reviewer 3's first point about the discrepancy between Figure 1 panels C and D, since Muskelin is one of the auto-ubiquitylated substrates of CTLH-Kdo, I would expect its levels to increase following Kdo KD, as shown in panel D, similar to what we often observe in the mammalian system. I understand the reviewer's concern that the data quality in panel C is not as clear, but Kdo DK 0-1 day shows generally lower loading based on Actin, indicating that this might be a WB quality issue. A better blot could address this matter.

We appreciate Reviewer #3's attention to detail and Reviewer #1 and #2's responses. We agree with Reviewer #2 that the intention of Figure 1C is to demonstrate that Muskelin is stabilized in the absence of Kdo or RanBPM while the intention of Figure 1E is to demonstrate a maintained interaction between ME31B and Muskelin in the absence of Kdo. We have included additional clarification in the text for the purpose of these experiments (lines 176-178; 191-194; 198-199). We also note that while the Muskelin protein levels decrease in the input lanes in Figure 1E, we disagree that this difference is "significant" and highlight that the levels remain the same between 2-3 hours and 4-5 hours with slightly varying Actin levels, as Reviewer #1 points out. The small discrepancies could be explained by slight differences in Western blotting between the experiments in 1C and 1E.

Indeed, in our replicates of the experiment in Figure 1E, we have found that Muskelin levels in the input lanes in the Kdo KD immunoprecipitations is consistent, as can be observed below. We opted to include the blots in Figure 1E because the interaction between Muskelin and ME31B is more interpretable than the blot below, where the signal from the Muskelin immunoprecipitation appears blown out and much more efficient relative to the GFP immunoprecipitation. However, we would be happy to switch the replicates used in the figure if the editor thinks this replicate would be less confusing.

We would also like to draw attention to Figure 5F, which is an independent experiment demonstrating that Muskelin is stabilized in 4-5 hour embryos lacking RanBPM. Thus, while the conclusion of Figure 1E aims to highlight the preserved interaction between ME31B and Muskelin in the absence of Kdo, we have shown in multiple experiments and with multiple experimental approaches that Muskelin is a target of the CTLH complex cascade and is stabilized in late embryos lacking CTLH complex components or Kdo.

Figure legend: The interaction between Muskelin and ME31B-GFP is more robust at the end of the MZT in the absence of Kdo. Muskelin or ME31B-GFP were immunoprecipitated from staged lysates across the MZT time course from control (mCherry) knockdown or Kdo knockdown embryos.

The reviewer is still concerned about the conclusion that Muskelin and ME31B may have a direct interaction. Since the pull-down assay failed to demonstrate this interaction, the authors should either employ other more effective methods to prove this conclusion or revise the conclusion section (Line 325) to ensure its accuracy, rather than merely discussing it.

Referee #2 response: From my perspective further experiments would be outside the scope of the current study. The authors could perhaps tone down the last sentence of the relevant section of the Results (Line 324-326) as "Together these data are consistent with the hypothesis that Muskelin serves as the *Drosophila* CTLH's substrate receptor for ME31B and, possibly, other components of the repressive complex".

Referee #1 response: Regarding the criticism of the direct or indirect interaction with MKLN1 and ME31b, I understand the reviewer's points and see that it could be a weakness of this paper. Based on our experience working on the MKLN1-containing CTLH complex, this complex is extremely challenging for biochemical assays using in vitro reconstitution. Just like the authors replied to reviewer 3, many missing/unidentified elements make it difficult to replicate what we observe in cells in vitro, even after purifying all the necessary components (at least 6 of them). To be honest, addressing what the reviewers are asking regarding the direct/indirect interaction and biochemical mechanism may not be possible to achieve in 1-2 years of intense research on it.

We respectfully disagree that the pull-down assay failed to demonstrate the interaction between Muskelin and ME31B and appreciate the suggestions from Reviewers #1 and #2. Indeed, in Figure 4, we demonstrate that Muskelin and ME31B exist within 12 Angstroms of one another by native crosslinking with DTSP followed by stringent denaturing washes, which we take with the rest of our data to indicate that Muskelin likely directly interacts with ME31B, although this may require other binding partners (a possibility we discuss). We have clarified our conclusion in the text as Reviewer #2 suggests (lines 336-338). In our attempts to perform further experimentation by purifying proteins, we have not yet been successful in purifying the entire CTLH complex and its targets, as Reviewer #1 mentions. We agree that these experiments are thus beyond the scope of the current manuscript.

This study found that Muskelin acts as a substrate adaptor of the highly regulated *Drosophila* embryonic CTLH E3 ligase in early embryos. Given that knocking down Muskelin has no apparent effect on embryo development, its function in early embryos remains a concern and requires further investigation.

Referee #2 response: This is unfair. The current study is at the molecular/biochemical not the developmental level. Effects on embryonic development are definitely outside the scope.

We agree that the finding that Muskelin depletion has no apparent effect on embryo development warrants further investigation. However, due to the many possibilities for why this effect is occurring, we agree with Reviewer #2 that these experiments would be outside the scope of the current manuscript.

Prof. Olivia Rissland
University of Colorado School of Medicine
United States

Dear Olivia,

I am very pleased to accept your manuscript for publication in the next available issue of EMBO reports. Thank you for your contribution to our journal.
